# The uncertainties in the laboratory-measured short-wave refractive indices of mineral dust aerosols and the derived optical properties: A theoretical assessment

Senyi Kong[1], Zheng Wang[1], Lei Bi[1]

[1]Key Laboratory of Geoscience Big Data and Deep Resource of Zhejiang Province, School of Earth Sciences, Zhejiang University, Hangzhou, 310027, China

*Correspondence to*: Lei Bi (bilei@zju.edu.cn)

**Abstract.** Mineral dust particles are nonspherical and inhomogeneous; however, they are often simplified as homogeneous spherical particles for retrieving the refractive indices from laboratory measurements of scattering and absorption coefficients. The retrieved refractive indices are then employed for computing the optical properties of spherical or nonspherical dust model particles with downstream applications. This study aims to theoretically investigate uncertainties involved in the aforementioned rationale based on numerical simulations and focuses on a wavelength range of 355 to 1064 nm. Initially, the optical properties of nonspherical and inhomogeneous dust aerosols are computed as baseline cases. Subsequently, the scattering and absorption coefficients of homogeneous spheres and super-spheroids are computed at various refractive indices and compared with those of inhomogeneous dust aerosols to determine the dust refractive index. To mimic the real laboratory measurement, the size distribution of the baseline case is assumed to be unknown and determined through a process akin to using optical particle counters for sizing. The resulting size distribution differs from the original one of the baseline cases. The impact of discrepancies in size distributions on retrieving the dust refractive index is also investigated. Our findings reveal that these discrepancies affect scattering and absorption coefficients, presenting challenges in accurately determining the refractive index, particularly for the real parts. Additionally, the retrieved refractive indices are noted to vary with particle size primarily due to differences in size distribution, with imaginary parts decreasing as the particle size increases. A comparison between sphere and super-spheroid models shows that the former tends to underestimate the imaginary parts, leading to an overestimation of single scattering albedo. This study underscores the importance of employing consistent nonspherical models for both refractive index retrieval and subsequent optical simulation in downstream applications. Nevertheless, the impact of refractive index uncertainties on asymmetry factor and phase matrix is found to be minimal, with particle shape playing a more significant role than differences in the imaginary parts of the dust refractive index.

## 1 Introduction

Dust aerosols, one of the most dominant aerosols globally, play a crucial role in regulating the energy budget of the Earth's climate system through direct, semi-direct, and indirect radiative effects (Ackerman et al., 2000; Atkinson et al., 2013; Hansen

et al., 1997; Kinne et al., 2006; Takemura et al., 2000). However, accurately describing the radiative effect of dust aerosols requires quantitative information about their shape, size distribution, and mineralogical composition, which still have large uncertainties (Adebiyi et al., 2023b; Di Biagio et al., 2020; Kok et al., 2023; Stegmann and Yang, 2017; Wang et al., 2020). Tremendous efforts have been devoted to developing nonspherical models for dust simulation including spheroids, tri-axial ellipsoids, super-spheroids, nonsymmetric hexahedra, Gaussian spheres and more realistic dust particle shapes (Bi et al., 2009,

2010, 2018a; Dubovik et al., 2006; Kahnert, 2015; Kalashnikova and Sokolik, 2004; Kemppinen et al., 2015a; Lin et al., 2018; Lindqvist et al., 2014; Mischenko et al., 1997; Saito et al., 2021). Among these models, the spheroid model is commonly used in remote sensing applications. To improve upon the spheroid model, the super-spheroid model, which extends the dimensions of both the sphere and spheroid models, has been proposed to provide a more comprehensive framework for describing the shape of dust particles. Initial studies have shown that homogeneous super-spheroid dust models align well with

laboratory measurements of scattering matrices and have demonstrated excellent performance in simulating polarized satellite measurements and airborne light detection and ranging (LiDAR) observations (Kong et al., 2022; Lin et al., 2018, 2021). To further enhance this research, inhomogeneous super-spheroid models have been developed using the newly updated invariant imbedding T-matrix (IITM) method (Wang et al., 2023).

The size distributions of dust aerosols from various sources under different conditions (transport or near sources) have been

widely investigated in a large number of laboratory measurements and field campaigns (Adebiyi et al., 2023a; Jeong, 2020; Kandler et al., 2009; Müller et al., 2011; Reid et al., 2003; Ryder et al., 2018; Tegen and Fung, 1994). However, an inherent ambiguity exists in the sizing of the dust particles (Reid et al., 2003). Most recently, Huang et al. (2021) reported that laboratory measurements by optical particle counters could underestimate the geometric diameter (volume-equivalent diameter) of dust aerosols with coarse sizes mainly due to the spherical assumption of dust particle shape.

Complex refractive indices (RI), which are fundamentally determined by the mineralogical composition of dust, play a critical role in calculating dust optical properties. The real parts ($n$) indicate the ratio of the speed of light in a vacuum to its speed in the medium, while the imaginary parts ($k$) denote the attenuation of light in the medium, which characterizes its absorptivity. It should be noted that the short-wave (355–1064 nm in this study) refractive indices of dust were normally retrieved from the extinction (or scattering) and absorption coefficients measured in the laboratory. In many retrievals, the heterogeneous and

irregular dust aerosols are simplified to homogeneous spherical particles, as calculating electromagnetic scattering by nonspherical particles is challenging, especially for large sizes (Di Biagio et al., 2017b, 2019; McConnell et al., 2010; Müller et al., 2009; Petzold et al., 2009; Ryder et al., 2013; Schladitz et al., 2009). However, ample evidence has demonstrated that assuming a spherical shape leads to significant biases in the optical properties of irregular dust aerosols (Bi et al., 2010; Castellanos et al., 2024; Dubovik et al., 2000; Mischenko et al., 2000; Nousiainen and Kandler, 2015). Therefore, it is

necessary to quantify the uncertainties associated with the resulting dust refractive indices obtained based on the assumption of homogeneous spherical particles.

Several studies have utilized spheroid models to obtain the refractive indices of irregular dust aerosols. Dubovik et al. (2000) examined the bias of such an assumption during retrieval of the optical properties for nonspherical dust particles from Aerosol

Robotic Network (AERONET) sun and sky radiance measurements. They considered realistic dust aerosols represented by spheroid models. However, the retrieval of the real parts of the refractive indices failed in most cases, while the imaginary parts could only be obtained with relatively large uncertainties under certain circumstances when using the Lorenz-Mie theory. Furthermore, when utilizing spheroid models for retrieval, achieving a good fit of the measured scattering matrices still required altering the shape combinations of spheroids, emphasizing the pivotal role of model shape in the retrieval of microphysical properties of nonspherical particles (Dubovik et al., 2006). Recently, spheroid models have been integrated into a sophisticated aerosol inversion algorithm known as the Generalized Retrieval of Aerosol and Surface Properties (GRASP). This algorithm facilitates the retrieval of refractive indices, size distributions, axis ratios, and vertical volume concentrations for dust and other aerosols (Dubovik et al., 2014, 2021). Besides, Veselovskii et al. (2010) conducted a comparison of retrievals of dust microphysical parameters using sphere and spheroid models. The findings revealed that the utilization of sphere models resulted in an underestimation of the real parts of the refractive indices and increased the uncertainties of other parameters. Bedareva et al. (2014) retrieved the microphysical and optical properties of dust aerosols from Sun-Sky radiance measurements using spheroids and compared them with the AERONET retrievals. The former mostly aligned with the AERONET results. However, the neglect of spectral variability in refractive indices led to an overestimation of both the real and imaginary parts of the refractive indices. Wagner et al. (2012) set the real parts of the refractive indices to 1.53 and retrieved the imaginary parts of Saharan soil samples from laboratory measurements of the extinction and absorption coefficients using a spheroid model. The derived imaginary parts were then used to evaluate the imaginary parts generated by the effective medium approximations. Similarly, Rocha-Lima et al. (2018) retrieved the imaginary parts of the fine-mode Saharan dust using the sphere and spheroid models at wavelengths of 350 to 2500 nm and assuming a constant value of 1.56 for the real parts. However, uncertainties in the refractive indices resulting from the model shapes were not resolved, as the results for two samples in that study were contradictory. In addition, there are still morphological differences between the spheroid models and the realistic nonspherical dust particles. For instance, Sorribas et al. (2015) used the spheroid models to simulate the scattering and backscattering coefficients of dust aerosols and compared them with the laboratory observations. While the spheroid models, in contrast to the sphere models, produced results that were closer to the observations, the computed scattering coefficients were nearly 49% smaller than the observed values. Kemppinen et al. (2015) investigated the reliability of the tri-axial ellipsoid models for retrieving the refractive indices from the scattering matrices, by considering the irregular inhomogeneous models as the actual dust particles. Nevertheless, a systematic quantification of uncertainties in laboratory measurements of dust refractive indices due to the spherical assumption is always lacking (Di Biagio et al., 2019).

Assessing such uncertainties of laboratory experiments of the dust samples is challenging. It is nearly impossible to find a model that precisely matches the morphology of the actual dust aerosols; thus, uncertainties regarding shape equivalence always exist. On the other hand, the true values of the optical properties of dust samples are still unknown. It is difficult to evaluate the extent to which the homogeneous models can accurately reproduce the true values using the retrieved refractive indices, as realistic dust particles are rarely homogeneous.

However, these uncertainties can be systematically investigated through numerical simulations. In this context, the inhomogeneous super-spheroid models can be considered as realistic representations of dust aerosols, serving as the baseline case with predefined microphysical properties. On the other hand, the homogeneous super-spheroid models, sharing identical shapes with the inhomogeneous counterparts, are used to illustrate an ideal scenario where inversion models match the shape of the target particles. Additionally, sphere models are used to emulate situations similar to the approach for retrieving refractive indices from laboratory experiments of optical properties. Within numerical simulations, all parameters can be adjusted, providing the advantage of discerning the sources of uncertainties and identifying the most influential factors that affect the results. The primary objective of this paper is to explore the implications of inversion models possessing the same shape as the target particles. Additionally, we aim to conduct a thorough theoretical examination of uncertainties that arise from principles in laboratory measurements of mineral dust refractive indices at short wavelengths. Furthermore, we investigate the consistency in optical properties between realistic dust aerosols and homogeneous models with retrieved refractive indices to examine the model performance.

This paper is organized in four sections. In Sect. 2, we describe the experimental design, including the overall procedure, the models and computational methods used, and the retrieval methods. Section 3 presents the results and discussions. The uncertainties of the dust refractive indices obtained in the laboratory, based on the assumption of homogeneous spherical particles, are investigated at different sizes and wavelengths. The corresponding optical properties are then calculated from the retrieved refractive indices and compared to those of the baseline case. Subsequently, we discuss the insights gained and how the uncertainties might manifest in a real laboratory setting. Finally, a summary is provided in Sect. 4.

## 2 Experimental design

### 2.1 Overall procedure

We conduct numerical simulations at five specific wavelengths (355, 532, 633, 865, 1064 nm) to assess the uncertainties in the dust refractive indices resulting from assuming a homogeneous spherical morphology. Note that, in this study, all results are obtained solely from numerical simulations, and no laboratory measurements are involved. Figure 1 illustrates a flowchart outlining the steps involved in the numerical simulations, which consists of four procedures:

1. The inhomogeneous super-spheroid dust models, internally mixed with several minerals (see Sect. 2.2), are considered as the baseline case, mimicking the dust samples used in the laboratory experiments. The homogenous sphere model and the homogeneous super-spheroid model (with the same shape as the inhomogeneous model) are used as the inversion models for retrieving the refractive indices. In this step, optical properties for inhomogeneous super-spheroid, homogeneous super-spheroid and sphere models at various sizes and wavelengths are calculated.

2. The size distribution of inhomogeneous super-spheroids as the baseline case are predefined, after which the absorption and scattering coefficients can be calculated accordingly.

3. The size distribution of the baseline case (with or without correction) is used to calculate the absorption and scattering coefficients of the inversion models (homogeneous super-spheroid and sphere models) for various refractive indices.

A look-up table of the absorption and scattering coefficients with respect to refractive indices is generated.

4. The refractive indices of the baseline case are retrieved. The absorption and scattering coefficients of the baseline case are located in the look-up table, and the corresponding refractive indices are determined. The Bouguer–Lambert method is also introduced for comparison.

In accordance with laboratory studies (Di Biagio et al., 2019), four instruments are considered in the numerical simulations:
Aethalometer, Nephelometer, scanning mobility particle sizer (SMPS) and Optical Particle Counter (OPC) (Figure 1; SMPS is not shown). In the numerical simulations, we have incorporated two correction processes based on the actual laboratory experiments. The first correction is the size correction, which is employed to determine the geometric size of the particles from imaginary OPC measurements. The OPC is typically used to measure the scattering intensity of individual particles and provides the diameter of the standard non-absorbing sphere model (polystyrene latex spheres, RI = 1.59+0i), which has
equivalent scattering intensity (Heim et al., 2008). However, the absorption of dust aerosols and the non-sphericity of the models (namely, super-spheroid) can introduce bias to the measured size distribution. Therefore, the size distribution measured using the OPC needs to be corrected before being used to establish a look-up table (Di Biagio et al., 2017; Huang et al., 2021; see Sect. 2.3.1). The second correction is the scattering truncation correction, which is associated with the unavoidable technical limitations in measurements of scattering coefficients. The Nephelometer measures the scattering coefficients
between 7 and $170^{o}$ due to difficulties in measurements in the forward $(0 - 7^{o})$ and backward $(170 - 180^{o})$ directions. Hence, a scattering truncation correction is needed to convert them into the scattering coefficients for the entire field of view $(0 - 180^{o}$; see Sect. 2.3.2). Four numerical simulations are designed to represent four scenarios of how the size distribution and target scattering coefficients for the inversion models differ from the baseline case (Table 1). Note that E1 represents an ideal situation in which the size distributions and scattering coefficients of the baseline case can be accurately obtained and
used in the retrieval, i.e., the inversion models share the identical size distribution with the baseline case, while E4 considers corresponding size and scattering truncation corrections and is the closest to the real laboratory experiments. The assumption made is that the absorption coefficients of the baseline case are obtained exactly in all simulations. Further explanations are provided in section 2.3.2.

**Table 1: A brief description of the four numerical simulations. Target scattering coefficient denotes the scattering coefficient of the**
**baseline case. See section 2.3 for more details.**

| | Target scattering coefficient used for inversion models | |
|---|---|---|
| Numerical simulations | the same as the baseline case | the scattering coefficients of the baseline case with scattering truncation correction |
| the same size as the baseline case | E1 | E2 |

| | Particle size of inversion models | the size of the baseline case with size correction | E3 | E4 |
|---|---|---|---|---|

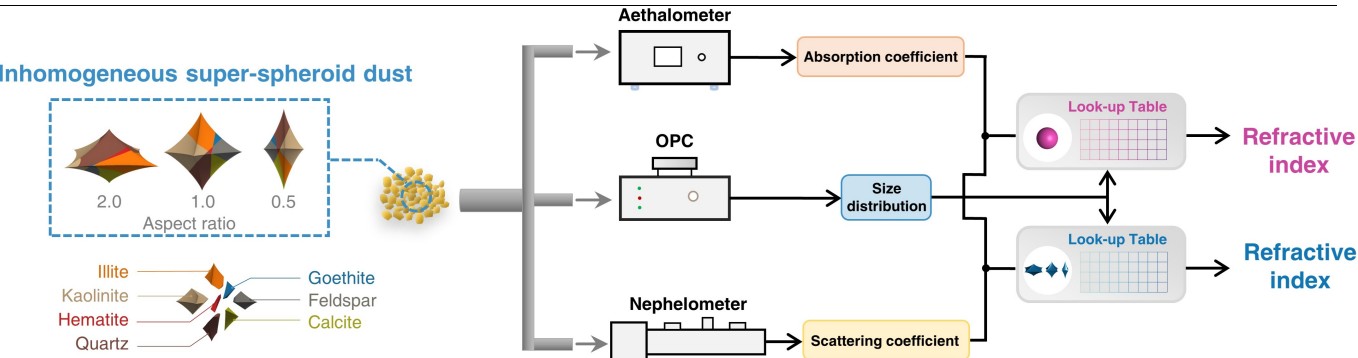

**Figure 1: A flowchart for the numerical simulations.**

## 2.2 Model and computational method

The super-spheroid models were developed for modelling atmospheric aerosols, including dust, sea salt, and ice crystals (Bi
et al., 2018b; Li et al., 2022; Lin et al., 2018; Sun et al., 2021). The equation for the super-spheroid model can be written as
shown below:

$$\left(\frac{x}{a}\right)^{2/e} + \left(\frac{y}{a}\right)^{2/e} + \left(\frac{z}{c}\right)^{2/e} = 1, \qquad (1)$$

in which a and c are the lengths of the semi-major axes along the corresponding coordinate axes, and e is the roundness
parameter. Specifically, a/c is defined as the aspect ratio. In this study, the value of e is fixed at 2.5, a value based on previous
studies (Kong et al., 2022; Lin et al., 2018, 2021). Additionally, three aspect ratios (0.5, 1.0, 2.0) are considered. Although we
have observed that equiprobable aspect ratios ranging from 0.5 to 2 are appropriate for fitting the measurements of dust aerosols
using the super-spheroid models (Lin et al., 2018), we restrict our selection to three aspect ratios (0.5, 1.0, 2.0) for the super-
spheroid models due to computational resource constraints. Nonetheless, models employing these chosen aspect ratios yield
optical properties comparable to those derived from models utilizing equiprobable aspect ratios. The mean values of their
optical properties are utilized to describe dust particles for both the inhomogeneous and homogeneous super-spheroid models.
The size parameter is defined as $\pi D_m/\lambda$, in which $D_m$ indicates the length of the longest axis of the particle, and $\lambda$ is the
wavelength. Optical properties are directly related to the size parameter instead of only the $D_m$. We calculate the single particle
optical properties using the IITM method for inhomogeneous super-spheroid models, considering random orientations (Bi et
al., 2013; Bi and Yang, 2014; Wang et al., 2023). The calculations are performed at size parameters ranging from 0.1 to 80
and wavelengths of 355, 532, 633, 865, and 1064 nm. Note that the maximum size parameter is extended to 100 for
wavelengths of 355 and 532 nm. This extension ensures that large particles at short wavelengths can be accurately
characterized. For homogeneous super-spheroid models, an optical database ranging from size parameter 0.1 to 1000 was

developed in previous studies (Yu et al., 2022). The IITM method was used for size parameters varying from 0.1 to 50, while the improved geometric optics method (IGOM) was applied for size parameters ranging from 50 to 1000 (Bi and Yang, 2017; Yang and Liou, 1996). Note that the IITM method is a rigorous algorithm, whereas the IGOM method is suited for the geometric-optics domain. However, the accuracy of the IGOM method and the effectiveness of combining these two methods were examined and validated in several studies (Bi et al., 2009; Bi and Yang, 2014; Lin et al., 2018; Yang et al., 2007). The uncertainties associated with the accuracy of optical properties are negligible compared to those of model assumptions in the numerical simulations. To reduce the computational burden, the neural network developed by Yu et al. (2022) was used. The single particle optical properties of sphere models are calculated using the Lorenz–Mie theory (Bohren and Huffman, 2008). In the optical database for homogeneous sphere and super-spheroid models, the real part of the refractive index ranges from 1.40 to 1.70 at intervals of 0.01, while the imaginary part varies from 0.0001 to 0.015 at steps of 0.0001. These values are determined based on literature values of refractive indices (Di Biagio et al., 2019 and references therein).

The baseline case refers to the inhomogeneous super-spheroid models (Wang et al., 2023; see Figure 1), chosen to represent the dust samples utilized in the laboratory experiments. This inhomogeneous model is based on previous evidence that suggests the presence of polymineralic aggregates in dust samples (Jeong and Nousiainen, 2014; Lindqvist et al., 2014). The mineralogy of dust samples from various sources is presented by Di Biagio et al. (2017). We choose the sample from Algeria for our study because it has a medium iron content, which closely approximates the mean values of the global average (Di Biagio et al., 2019; Go et al., 2022). Previous evidence has reported that the mineralogical composition varies with particle sizes (Kandler et al., 2007, 2009). However, for the purposes of this study, we assume that the composition is size-independent in order to simplify the scientific question and investigate the effects of morphology. The mineral composition consists of 45.1% kaolinite, 21.5% quartz, 18.3% illite, 7.9% feldspar, 4.4% calcite, 1.4% goethite, and 1.4% hematite by mass concentration.

The refractive indices of several major mineral components are shown in Figure 2. It is evident that hematite and goethite exhibit the highest absorption at short wavelengths. The clays, including chlorite, illite, kaolinite, and montmorillonite, have similar refractive indices due to their similar chemical compositions. Quartz, on the other hand, has negligible absorption. The refractive indices of goethite, as reported by Bedidi and Cervelle (1993), are only available for wavelengths ranging from 460 to 700 nm. Unfortunately, no other reported values are available for reference. It is worth noting that the refractive indices of goethite are similar to those of magnetite, particularly in terms of the imaginary parts. Therefore, we adopt the refractive indices of magnetite as an alternative for goethite for wavelengths below 460 nm and above 700 nm. The current reported values of hematite refractive indices have large uncertainties (Go et al., 2022). However, the general trends and magnitudes among these refractive indices at the five selected wavelengths are relatively close. In our numerical simulations, we assume the refractive indices of various minerals used in this study to be accurate. It is important to mention that these numerical simulations are conducted theoretically and did not involve any actual measurement values. Therefore, the accuracy of the refractive indices of the minerals had little influence on our analysis.

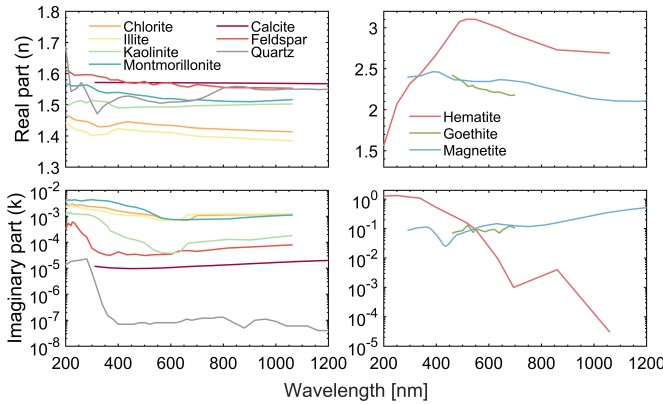


**Figure 2: The refractive indices (RI) of various minerals at the wavelengths of 200 to 1200 nm. The RIs of calcite are obtained from the work of Roush (2021); The RIs of chlorite are obtained from the work of Lee et al. (2020); the RIs of feldspar, illite, kaolinite, and montmorillonite are obtained from the work of Egan and Hilgeman (1979); The RIs of hematite are obtained from the work of Longtin et al. (1988); The RIs of quartz are obtained from the work of Khashan and Nassif (2001); The RIs of goethite**

**are obtained from the work of Bedidi and Cervelle (1993), and the RIs of magnetite are obtained from the work of Querry (1987).**

## 2.3 Correction processes

### 2.3.1 Size distribution

In the numerical simulations, the size distributions of inversion models are derived following the method described by Di Biagio et al. (2019). Assuming that the OPC is used to measure the scattering intensity from individual particles at specific

wavelengths within a defined range of scattering angles ($\theta_{min} - \theta_{max}$), which can be expressed as shown below:

$$I_{sca} = \tfrac{1}{2}I_0 C_{sca} \int_{\theta_{min}}^{\theta_{max}} P(\theta)\sin\theta \mathrm{d}\theta. \tag{2}$$

$I_{sca}$ represents the scattering intensity, $I_0$ represents the incident intensity of the OPC light source, $C_{sca}$ represents the scattering cross section, $P(\theta)$ represents the phase function, and θ denotes the scattering angle. We consider two types of OPC. One is the skyGrimm OPC (referred to as GRIMM), which operates at a wavelength of 655 nm and covers an angular range

from $30^o$ to $150^o$ (Bundke et al., 2015). The other one is the WELAS OPC (referred to as WELAS), which uses a 4200K white light Xenon arc lamp and a $90^o$ scattering angle (Heim et al., 2008). Given a specific model, the scattering intensity is tabulated as a function of the size parameter of a single particle theoretically, and then the size of a realistic particle can be determined once the scattering intensity is measured. Normally, the diameter is derived from the intensity measured by the OPC based on a sphere model with the refractive index of polystyrene latex (RI = 1.59+0i), and this diameter is referred to as

the optical diameter ($D_{opt}$). In reality, the refractive index could be different and the particle could be nonspherical and inhomogeneous. The conversions from the aforementioned optical diameter ($D_{opt}$) derived from the OPC to the geometric diameter ($D_{geo}$) of various models should be conducted. Prior values of refractive indices are needed for the conversions. In this study, the real parts of the refractive indices ($n$) for the homogeneous models are set to 1.47, 1.50, and 1.53, while the imaginary parts ($k$) vary from 0.001 to 0.005 in 0.001 increments based on the values provided by Di Biagio et al. (2019). Note

that, the SMPS is used for $D_{\text{geo}}$ values smaller than 0.3 µm, the GRIMM is used for 0.3 µm $< D_{\text{geo}} \leq 1$ µm, and the WELAS is used for $D_{\text{geo}} > 1$ µm (Di Biagio et al., 2019). The $D_{\text{geo}}$ obtained from the SMPS does not need to be converted among the models with different shapes because the detection values are directly related to the particle mass instead of optical properties. Therefore, the conversion results for the GRIMM and WELAS measurements are obtained and shown in Figure 3a, b, respectively. The results corresponding to different models are indicated in the figure. The geometric diameter of super-spheroids represents the diameter of a volume-equivalent sphere. We found that the conversions are different for the GRIMM and WELAS due to the differences in the range of scattering angles and the wavelength of the light source. The significance of the differences in the phase function decreases when integrated over a wide angular range (Mishchenko et al., 1997). Hence, the conversions for the sphere and super-spheroid models are more similar for the GRIMM than for the WELAS. Nevertheless, the $D_{\text{geo}}$ for the super-spheroid models are smaller than those for the sphere models.

The conversions between the $D_{\text{geo}}$ for the sphere models and that for the super-spheroid models are illustrated in Figure 3c, d. Note that the $D_{\text{geo}}$ for the sphere models are nearly comparable to the $D_{\text{geo}}$ for both the inhomogeneous and homogeneous super-spheroid models when $D_{\text{geo}}$ is smaller than 1 µm. However, the $D_{\text{geo}}$ of the super-spheroid models are significantly overestimated when assuming the sphere models for relatively large sizes. This overestimation is also indicated by Huang et al. (2021). The conversion factors ($C_{\text{f}}$), defined as $D_{\text{geo,super-spheroid}}/D_{\text{geo,sphere}}$, are found to be smaller than 0.5 at 10 µm. The $C_{\text{f}}$ values of the inhomogeneous models are about 8% lower than those of the homogeneous models. This suggests that the bias in retrieving size distribution for inhomogeneous and irregular particles is not only caused by the difference in model shapes, but also by the imperfect representation of inhomogeneity. According to Eq. (2), any disparity in the microphysical properties (e.g., shape, absorptivity) will result in a difference in $I_{\text{sca}}$, ultimately leading to a bias in $C_{\text{f}}$ values. However, the variation in model shapes leads to a dominant bias when using sphere models for retrieval. The use of incorrect refractive indices for the conversion of size can introduce biases in the converted particle size when comparing homogeneous and inhomogeneous models. Prior values of refractive indices are crucial for accurate size conversion. Sensitivity studies have shown that the conversions are much less sensitive to $n$ than to $k$. The change in absorptivity results in significant variations in the scattering cross section, as dust aerosols exhibit moderate absorption in the visible band. The first guess values of $k$ is essential. We consider the literature values in this study (Di Biagio et al., 2019). However, the literature value is not the sole option, the first guess values of refractive indices can be optimized through an iterative retrieval. The retrieval can start with a rough guess value and then obtain a more precise value. The retrieved values can be used as new prior values, and the retrieval can be repeated until optimal values are obtained. To maintain consistency with the laboratory studies (Di Biagio et al., 2017b, 2019; Ryder et al., 2013), we do not consider this iterative retrieval to obtain the size distribution. The conversion factors for homogeneous models and inhomogeneous models are typically similar if they have comparable absorptivity. For instance, the inhomogeneous super-spheroid models showed similar trends to the low-absorbing homogeneous super-spheroid models in which $k = 0.001$ (Figure 3a, b).

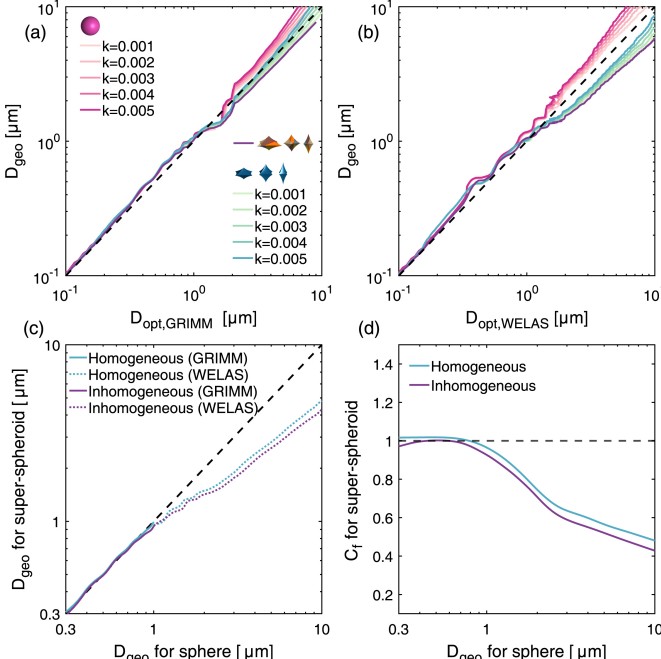

**Figure 3: The conversions among the optical diameters ($D_{opt}$) obtained by the optical particle counters of GRIMM and WELAS, the geometrical diameters ($D_{geo}$) obtained from the homogeneous sphere models, the homogeneous super-spheroid models and the inhomogeneous super-spheroid models. Panels (a) and (b) display the geometrical diameters ($D_{geo}$) obtained from different models and compared to the optical diameters measured by the optical particle counters of GRIMM and WELAS. The real part ($n$) of the refractive index for the homogeneous models is 1.50 and the imaginary part ($k$) varies from 0.001 to 0.005. Panel (c) illustrates the $D_{geo}$ conversions from the sphere models to the homogeneous and inhomogeneous super-spheroid models, while Panel (d) shows the corresponding conversion factors ($C_f$). The mean values of the conversion results are illustrated for which $n$ = 1.47, 1.50, 1.53 and $k$ = 0.001, 0.002, 0.003, 0.004, and 0.005 are chosen for the homogeneous models.**

The particle number size distributions in the four numerical simulations are adopted from the five-modal lognormal size distributions of the Algerian dust samples reported by Di Biagio et al. (2019) and displayed in Figure 4 (corresponding to the sphere case in Fig. 4b). To closely resemble actual laboratory conditions, the size distributions for sphere models in E3 and E4 are initially set to be this reported size distribution (Di Biagio et al., 2019). For sensitivity studies, we specifically consider three size distributions representing small (S), medium (M), and large (L) particles, respectively. The size distribution (S) represents mode 1, the size distribution (M) includes modes 1–3, and the size distribution (L) consists of all five modes (Table 2).

The size conversions between the sphere models and the super-spheroid models can be expressed as shown below:

$$\frac{dN}{dD_{geo}} = \frac{dN}{dD_{geo,sphere}} \cdot \frac{dD_{geo,sphere}}{dD_{geo}} = \frac{dN}{dD_{geo,sphere}} \cdot \left( \frac{1}{C_f} - \frac{D_{geo} \cdot \frac{dC_f}{dD_{geo}}}{C_f^{\,2}} \right), \tag{3}$$

in which $D_{geo} = D_{geo,sphere} \cdot C_f$, representing the geometrical diameter for the inhomogeneous or homogeneous super-spheroid models. Then, the size distributions for the super-spheroid models ($dN/dD_{geo}$) (in Figure 4b) are derived from the

size distributions for the sphere models ($dN/dD_\text{geo,sphere}$) using Eq. (3). For a specific size parameter, the $D_\text{geo}$ for super-spheroids are smaller than those for spheres.

Different from E3/E4 scenarios, the size distribution for all models (sphere, homogeneous and inhomogeneous super-spheroids) in E1 and E2 are assumed to the same. The size distribution in E1 and E2 are the same as their counterparts of spheres in E3 and E4, except the size distribution at the large size (L), which is modified to be smaller than that in E3 and E4. This is done to ensure that all databases of the various models can encompass 99.9% of the cumulative distribution function of the volume size distributions. This approach is taken because the focus of this study is not to compare different numerical simulations, but rather to examine the uncertainties within each simulation.

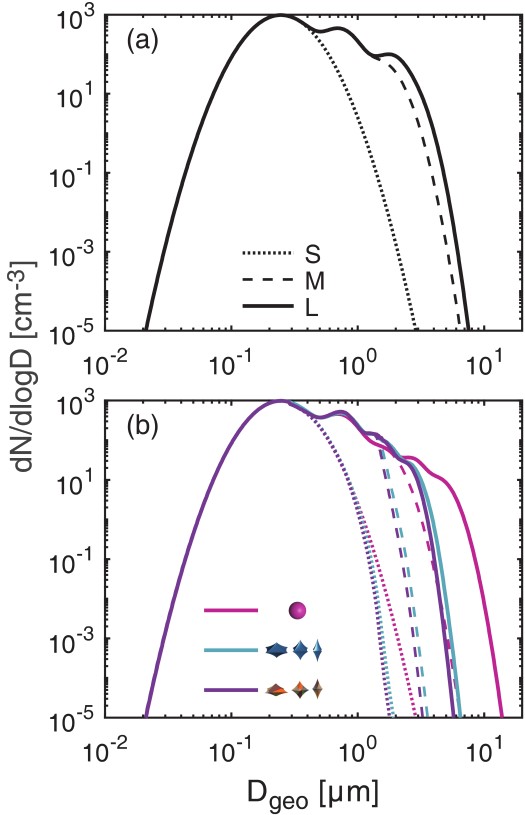

Figure 4: The size distributions in E1/E2 (a) and E3/E4 (b).

Table 2: The parameters for the five-modal lognormal size distributions for sphere models in the numerical simulations. $N$ indicates the number of concentrations (unit: $\text{cm}^{-3}$), $D_\text{g}$ represents the geometric mean (unit: µm), and $\sigma_\text{g}$ denotes the geometric standard deviation (unitless).

| Numerical | Mode 1 | | | Mode 2 | | | Mode 3 | | | Mode 4 | | | Mode 5 | | |
|---|---|---|---|---|---|---|---|---|---|---|---|---|---|---|---|
| simulations | $N$ | $D_\text{g}$ | $\sigma_\text{g}$ | $N$ | $D_\text{g}$ | $\sigma_\text{g}$ | $N$ | $D_\text{g}$ | $\sigma_\text{g}$ | $N$ | $D_\text{g}$ | $\sigma_\text{g}$ | $N$ | $D_\text{g}$ | $\sigma_\text{g}$ |
| E1/E2 | 267 | 0.29 | 1.50 | 207 | 0.77 | 1.30 | 65 | 1.60 | 1.30 | 37 | 1.96 | 1.20 | 26 | 2.48 | 1.24 |

| E3/E4 | 267 | 0.29 | 1.50 | 207 | 0.77 | 1.30 | 65 | 1.60 | 1.30 | 37 | 2.80 | 1.20 | 26 | 4.50 | 1.25 |


### 2.3.2 Scattering and absorption coefficients

The scattering and absorption coefficients are obtained using the Nephelometer (TSI Inc. model 3563) and Aethalometer (Magee Sci. AE31 model), respectively. An angular truncation exists in the Nephelometer, which can only be used to measure the scattering coefficients between 7 and $170^o$ ($\beta_{\text{sca}}(\theta_7 - \theta_{170})$) due to the limits of the instrument. To obtain the scattering

coefficients for the entire field of view ($\beta_{\text{sca}}(\theta_0 - \theta_{180})$), a scattering truncation correction is needed. The scattering coefficients for an angular range from $\theta_{\text{min}}$ to $\theta_{\text{max}}$ can be expressed as shown below:

$$\beta_{\text{sca}}(\theta_{\text{min}} - \theta_{\text{max}}) = \frac{1}{2}\int_{D_{\text{geo,min}}}^{D_{\text{geo,max}}} \int_{\theta_{\text{min}}}^{\theta_{\text{max}}} P(D_{\text{geo}}, \theta) \cdot \sin\theta \cdot C_{\text{sca}}(D_{\text{geo}}) \cdot \frac{\text{d}N}{\text{d}D_{\text{geo}}} \text{d}\theta \text{d}D_{\text{geo}}. \tag{4}$$

Hence, $\beta_{\text{sca}}(\theta_0 - \theta_{180})$ can be easily obtained by multiplying $\beta_{\text{sca}}(\theta_7 - \theta_{170})$ by the truncation factor ($C_{\text{trunc}}$). $C_{\text{trunc}}$ is calculated based on the equation shown below:

$$C_{\text{trunc}} = \beta_{\text{sca,model}}(\theta_0 - \theta_{180})/\beta_{\text{sca,model}}(\theta_7 - \theta_{170}). \tag{5}$$

In E1 and E3, we make the assumption that the $\beta_{\text{sca}}(\theta_0 - \theta_{180})$ can be directly obtained by the instrument. However, in E2 and E4, the $\beta_{\text{sca}}(\theta_0 - \theta_{180})$ is corrected from $\beta_{\text{sca}}(\theta_7 - \theta_{170})$. Prior values of the refractive indices are also needed for calculating the $C_{\text{trunc}}$. The values of $n$ are fixed at 1.53, while $k$ is set to 0.004, 0.003, 0.001, 0.002, and 0.003 at 355, 532, 633, 865, and 1064 nm, respectively. These values are adopted from the study by Di Biagio et al. (2019) and modified based on the absorptivity of the baseline case. Sensitivity tests show that a variation of 0.001 in $k$ resulted in a 0.4% variation in $C_{\text{trunc}}$,

while a variation of 0.03 in $n$ resulted in only a 0.1% variation in $C_{\text{trunc}}$. Thus, it is believed that the uncertainties caused by the prior values of the refractive indices are small. Generally, $C_{\text{trunc}}$ varies in the range of 1.1 to 1.7, and increases with size but decreases with wavelength. The differences in $C_{\text{trunc}}$ between the super-spheroid models and the sphere models are approximately 1.5% at the large size (L) in E1 and E2. Such small differences are also reported by Sorribas et al. (2015). It is

reasonable to observe such a difference because the influence of shape on the phase function is less significant when integrated over the size distribution. However, such differences are large (up to approximately 25%) at large sizes in E3 and E4 due to the large differences in the corrected size distributions.

Actually, the Aethalometer (Magee Sci. AE31 model) did not directly measure absorption coefficients but rather attenuation coefficients (Hansen et al., 1984). Extensive efforts have been made to determine the absorption coefficients by accounting

for corrections related to scattering, loading, and multiple scattering effects, using the attenuation and scattering coefficients (Arnott et al., 2005; Collaud Coen et al., 2010; Di Biagio et al., 2017a; Schmid et al., 2006; Virkkula et al., 2007; Weingartner et al., 2003). Given that the corrections for dust particles are conducted using empirical formulas and are challenging to verify through numerical simulations. In addition, the corrections have been validated using measurements from the Multi-Angle Absorption Photometer (MAAP) and Cavity Attenuated Phase Shift Extinction (CAPS) (Di Biagio et al., 2017a). As a result,

we assume that the absorption coefficient can be accurately obtained within a certain uncertainty, and thus, no additional correction is applied in the numerical simulations.

## 2.4 Retrieval method

### 2.4.1 Look-up table: exact and range values

The root mean square deviation ($RMSD$) of the scattering coefficients and the absorption coefficients are calculated for various
refractive indices in the look-up table at each wavelength and size, following the methods in previous studies (Di Biagio et al., 2019; Wagner et al., 2012). The formula can be expressed as shown below:

$$RMSD(X,\lambda,n,k) = \sqrt{\left(\frac{\beta_{sca}(X,\lambda,n,k) - \beta_{sca,model}(X,\lambda,n,k)}{\beta_{sca,model}(X,\lambda,n,k)}\right)^2 + \left(\frac{\beta_{abs}(X,\lambda,n,k) - \beta_{abs,model}(X,\lambda,n,k)}{\beta_{abs,model}(X,\lambda,n,k)}\right)^2}. \quad (6)$$

The variable $X$ represents the size distribution and can be either S, M, or L. The minimum value of $RMSD$ indicates the refractive indices with the best agreement. However, due to the sparse nature of the look-up table, it is highly unlikely for the
target value to fall directly on the grid points. Therefore, we average the four refractive indices corresponding to the four smallest values of $RMSD$. These average refractive indices are referred to as the exact values.

Di Biagio et al. (2019) provided an estimation of the uncertainty in the scattering and absorption coefficients. They found that the relative uncertainty in the scattering coefficients ranged 5% to 12%, while, for the absorption coefficients, it ranged from 22% to 30% at 370 nm and 23% to 87% at 950 nm. In this study, we assume a relative uncertainty of 8% in the scattering
coefficients and 30% in the absorption coefficients. By considering these uncertainties, we are able to obtain the range of possible refractive indices. However, if the target values are not covered within the range of the look-up table, the retrieved refractive indices are discarded. It is worth noting that the target absorption coefficients are always within the range of the look-up table, as shown in Figure 5. Therefore, we also demonstrate the potential range of imaginary parts by solely considering the absorption coefficients. In this study, we do not consider the Kramers–Kronig relationship between $n$ and $k$, as we only
obtain the refractive indices at five wavelengths.

### 2.4.2 Bouguer–Lambert method

The Bouguer–Lambert method was frequently used in earlier studies (Patterson et al., 1977; Sokolik et al., 1993; Volz, 1972) to determine $k$ based on the absorption coefficient. By considering the space containing dust aerosols and air as a homogeneous medium, the value of $k$ can be derived using the equation shown below:

$$k = \frac{\lambda\beta_{abs,medium}}{4\pi}, \quad (7)$$

in which $\beta_{abs,medium} = \beta_{abs}/V_{dust}$, and $V_{dust}$ denotes the volume of the ensemble of dust particles. The advantage of this method is that it eliminates the need for any optical calculations. However, a disadvantage is that it may not provide accurate results due to unrealistic assumptions. Nonetheless, this method can still be used for comparison purposes.

## 3 Results and discussion

### 3.1 The retrieved refractive indices in E1/E2

Figure 5 illustrates the scattering and absorption coefficients of the baseline case (target values), as well as the look-up tables for the super-spheroid models and sphere models in the simulation scenarios of E1 and E2. It is noteworthy that the overall dimensions of the look-up tables diminish with increasing size. As the particle size increases, the scattering coefficients become less sensitive to changes in the real parts of the refractive indices. This phenomenon can be explained by the optical theorem, which states that the extinction cross section(including scattering and absorption cross sections) are approximately twice the geometric projected area as the size increases, regardless of the refractive indices (Liou, 2002). However, the absorption coefficients are significantly influenced by the imaginary parts, which in turn affect the scattering coefficients. At small sizes (Figure 5c), both the look-up tables for the super-spheroid and sphere models cover the target values, while the sphere model's range barely matches the target values at large sizes (Figure 5a). This finding is consistent with previous studies that have shown a large discrepancy between the measured scattering coefficients and the calculated counterparts due to the non-sphericity of large particles (Schladitz et al., 2009). Therefore, $n$ is typically fixed at a specific value, such as 1.53, in the retrieval of the refractive indices (Müller et al., 2009; Schladitz et al., 2009; Wagner et al., 2012). However, the target absorption coefficients are always within the range of both look-up tables. The influence of the scattering truncation correction on the scattering coefficients increases with size. For a small size, the correction is nearly negligible. Note that the exact target values exceed the range of the look-up tables for the super-spheroid models at large sizes and at wavelengths of 355 and 532 nm when considering such a correction. The uncertainty range of the scattering coefficients is large compared to the range of the look-up tables. At a large size, $n$ can vary from 1.40 to 1.70 within the uncertainty. Accurately retrieving $n$ is challenging. However, in an ideal scenario, the target values can fall within the range of the look-up tables, and the corresponding refractive indices can be retrieved at any size, when the baseline case and the inversion models share identical size distribution and shape. Note that the ambiguous definition of size for irregular particles could also lead to discrepancies between the measurements and the simulations based on the sphere models (Chen et al., 2011). Saito and Yang (2022) suggested that the effective radius, defined as three times the volume divided by four times the average projected area, was the most appropriate size descriptor for non-spherical particles. However, we find that the discrepancies are even larger when the effective radius is used. The effective radius is smaller than the geometric radius at the same size parameter for the super-spheroid model. As a result, the simulated scattering coefficients of the sphere models using the effective radius are smaller than those using the geometric radius. The retrieval fails even at small size (S), and the difference of scattering coefficients between those calculated by sphere models and the baseline case can be 50-70% depending on the size and wavelength. Therefore, it is believed that the geometric diameter is better suited for retrieval in this study.

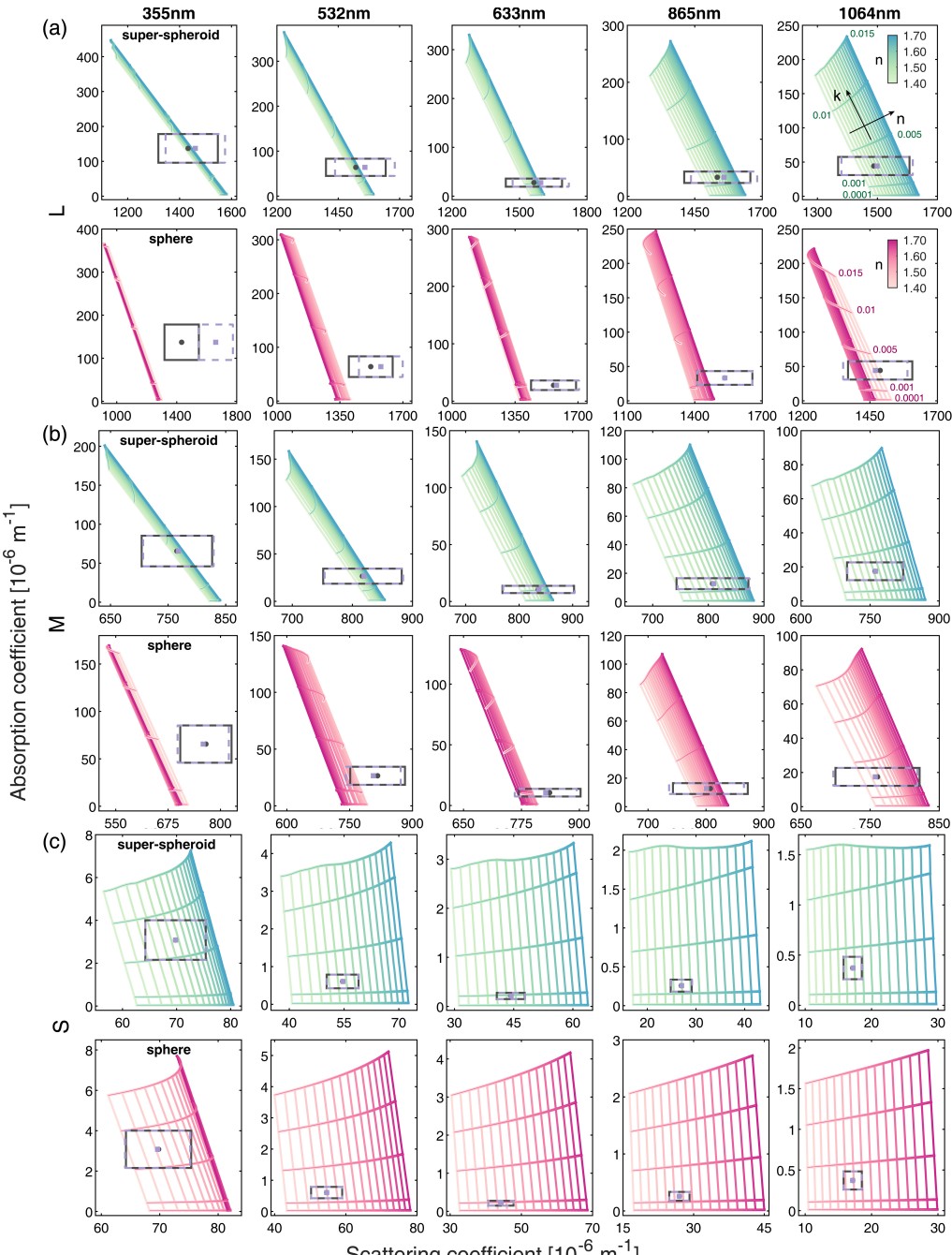

**Figure 5:** The look-up tables for refractive indices produced by the sphere model and super-spheroid model with different particle sizes (S, M, L in (a), (b), and (c), respectively) to determine the absorption coefficient versus the scattering coefficient for E1/E2. The black round point and rectangle denote the exact target value and its corresponding uncertainties whereas the light purple square point and rectangle with dashed line denote the target value and its corresponding uncertainties after truncation correction. In the look-up table, the real parts (*n*) are represented by different colors, and five values of the imaginary parts (*k*) (0.0001, 0.001, 0.005, 0.01, and 0.015) are displayed.

The target absorption coefficients exhibit a decrease with increasing wavelength until reaching a minimum value at a wavelength of 633 nm, beyond which they increase (Figure 6a). Consequently, this trend is also observed in the imaginary parts of the refractive index, which displays a similar bow-shaped signature (Figure 6c). This behaviour can be attributed to the imaginary parts of goethite. Note that we adopt the refractive indices of magnetite as an alternative for goethite at wavelengths of above 700 nm due to the lack of direct measurements. At these longer wavelengths, the absorptivity of dust is mainly determined by goethite, as hematite exhibits weak absorption (Go et al., 2022). Hence, the absorptivity increases with wavelengths above 633 nm.

Rocha-Lima et al. (2018) conducted a study on Saharan dust and derived $k$ across a range of wavelengths from 350 to 2500 nm. Their findings also showed a bow-shaped signature of $k$, with a minimum value observed at approximately 650 nm for fine-mode particles. However, the increasing trend of the absorption coefficients ranging from 633 to 1064 nm and the bow-shaped signature of $k$ were not consistently observed in many actual laboratory measurements (Di Biagio et al., 2019; Müller et al., 2009, 2011; Wagner et al., 2012). While some studies indeed have demonstrated a bow-shaped signature of $k$, but the absorptivity of dust either weakened or remained unchanged below 1064 nm (Balkanski et al., 2007; Wells et al., 2012). Therefore, it is possible that the imaginary parts of goethite may have been overestimated above 700 nm by assuming the $k$ of magnetite. Accurate measurement of the refractive indices of goethite at shorter wavelengths are still required.

Given that the retrieved refractive indices in E1 and E2 are similar, and the differences can be deduced from Figure 5, we only display the results for E1 in Table 3. Note that in Figure 6b and c, the exact retrieved refractive indices for the super-spheroid models are available at all sizes, while those for the sphere models are only available at a small size and partly available at a medium size. In cases where target values exceed the range of the look-up table, "nan" values are provided in Table 3 (the same in Table 4). Generally, no obvious regulations are found in the real parts at different sizes but a clear decreasing trend with size is observed for the imaginary parts. It can be deduced that simulating the optical properties of the inhomogeneous models using a homogeneous model and a single refractive index is nearly impossible, as the homogeneous models can not accurately represent them. The phenomenon that different refractive indices may be obtained for particles of varying sizes is also noted in the laboratory measurements (Orofino et al., 1998).

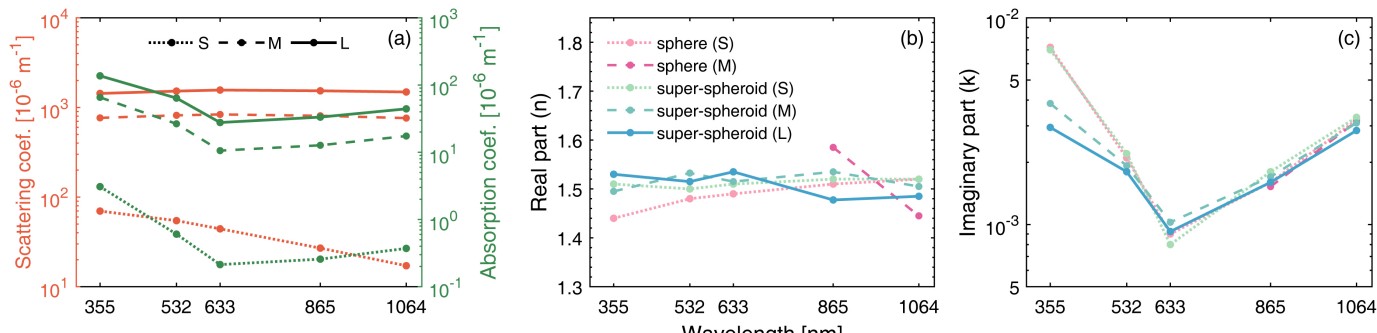

**Figure 6**: The scattering coefficients and absorption coefficients of the baseline case (a) and the exact refractive indices retrieved from the homogeneous sphere models and the homogeneous super-spheroid models (b, c) for different particle sizes (S, M, L) for E1.

Figure 7 displays the range values of $k$ for the super-spheroid and sphere models, as well as their differences in E1. The $k$
obtained through the Bouguer–Lambert method is not influenced by the models in E1, as the volume size distributions are the
same for all the models. Note that the $k$ obtained through the Bouguer-Lambert method are substantially larger than those
retrieved by the look-up table. This finding can be attributed to the unrealistic assumption made in the Bouguer–Lambert
method. Similar to the exact values in Figure 6b and c, $k$ generally decreases with size, but the trend is less significant under
weak-absorption conditions. The $k$ retrieved by the sphere models are close to those retrieved by the super-spheroid models.
The differences are relatively higher under strong-absorption conditions and reach up to 0.0006, while they are less than 0.0001
under weak-absorption conditions. Therefore, retrieving the imaginary parts solely from the absorption coefficients exhibits
reduced sensitivity to the model shape when identical size distributions are utilized. This is because the extinction coefficients
are primarily influenced by particle sizes, and the calculated absorption coefficients are similar for models with the same size
distributions. As a result, the target absorption coefficient is mapped to similar imaginary parts in the look-up tables for
inversion models with the same size distribution.

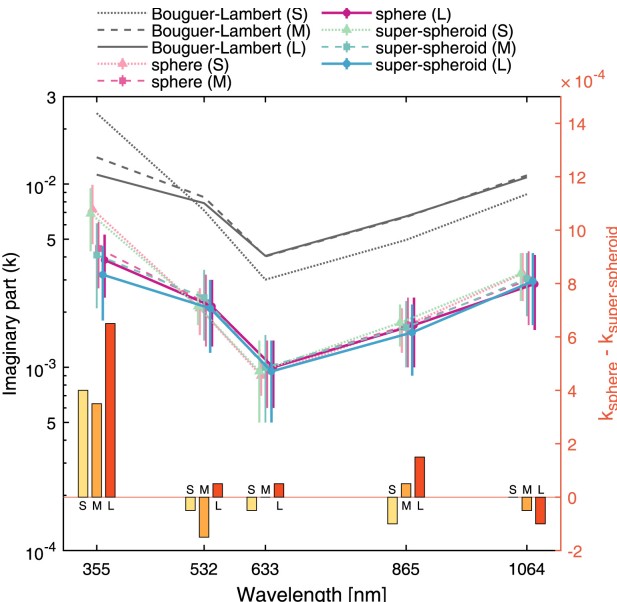

**Figure 7: The wavelength-dependent imaginary parts ($k$) of refractive indices obtained from absorption coefficients (left y-axis) and the differences in $k$ retrieved from the homogeneous sphere models and the homogeneous super-spheroid models (right y-axis) for different particle sizes (S, M, L) for E1/E2. The imaginary parts retrieved by the Bouguer–Lambert method are included for comparison. The error bar indicates half of the range of $k$, and the marker represents the mean of the range. To clarify the uncertainty of $k$, the data points are slightly shifted horizontally.**

Table 3: The refractive indices retrieved from the homogeneous sphere models and the homogeneous super-spheroid models at different sizes (S, M, L) at the wavelengths of 355 to 1064 nm for E1. "exact" indicates the exact refractive while "range" represents the range values as the uncertainties are considered. The "nan" values indicate that target values fall outside the look-up table.

| Numerical simulation E1 | | | 355 nm | | 532 nm | | 633 nm | | 865 nm | | 1064 nm | |
|---|---|---|---|---|---|---|---|---|---|---|---|---|
| | | | n | k | n | k | n | k | n | k | n | k |
| S | sphere | exact | 1.44 | 0.0072 | 1.48 | 0.0021 | 1.49 | 0.0009 | 1.51 | 0.0016 | 1.52 | 0.0032 |
| | | range | 1.40-1.53 | 0.0047-0.0099 | 1.45-1.51 | 0.0015-0.0027 | 1.47-1.51 | 0.0007-0.0011 | 1.49-1.53 | 0.0012-0.0021 | 1.50-1.53 | 0.0023-0.0042 |
| | super-spheroid | exact | 1.51 | 0.0070 | 1.50 | 0.0022 | 1.51 | 0.0008 | 1.52 | 0.0018 | 1.52 | 0.0033 |
| | | range | 1.44-1.65 | 0.0043-0.0095 | 1.48-1.55 | 0.0017-0.0026 | 1.48-1.54 | 0.0005-0.0014 | 1.50-1.54 | 0.0013-0.0022 | 1.50-1.53 | 0.0023-0.0042 |
| M | sphere | exact | nan | nan | nan | nan | nan | nan | 1.59 | 0.0015 | 1.45 | 0.0032 |
| | | range | nan | 0.0027-0.0062 | nan | 0.0013-0.0032 | nan | 0.0006-0.0014 | 1.40-1.70 | 0.0010-0.0024 | 1.40-1.70 | 0.0017-0.0043 |
| | super-spheroid | exact | 1.50 | 0.0039 | 1.53 | 0.0019 | 1.52 | 0.0010 | 1.54 | 0.0017 | 1.51 | 0.0031 |
| | | range | 1.40-1.70 | 0.0021-0.0061 | 1.40-1.70 | 0.0014-0.0034 | 1.40-1.70 | 0.0005-0.0015 | 1.40-1.70 | 0.0010-0.0023 | 1.42-1.63 | 0.0019-0.0042 |
| L | sphere | exact | nan | nan | nan | nan | nan | nan | nan | nan | nan | nan |
| | | range | nan | 0.0024-0.0053 | nan | 0.0013-0.0030 | nan | 0.0006-0.0014 | nan | 0.0010-0.0024 | nan | 0.0016-0.0041 |
| | super-spheroid | exact | 1.53 | 0.0030 | 1.52 | 0.0018 | 1.54 | 0.0010 | 1.48 | 0.0016 | 1.49 | 0.0029 |
| | | range | 1.40-1.70 | 0.0018-0.0046 | 1.40-1.70 | 0.0012-0.0030 | 1.40-1.70 | 0.0005-0.0014 | 1.40-1.70 | 0.0009-0.0022 | 1.40-1.70 | 0.0017-0.0042 |

## 3.2 Retrieved refractive indices in E3/E4

Similar to Figure 5, the target values and the look-up tables in E3 and E4 are illustrated in Figure 8. Note that a significant discrepancy emerges between the baseline case and the homogeneous super-spheroid models as the size increases, which is inconsistent with the findings in Figure 5. Furthermore, the discrepancy for the sphere models is even larger. This discrepancy can be attributed to the differences in size distributions. These differences are not influenced by the size descriptor of the non-spherical particle but are directly caused by the discrepancies in the optical properties between the baseline case and the models when using the OPC to measure the size of individual particles.

As described in Eq. (2), the OPC measures the scattering intensity of individual particles with a metric that is influenced by both particle size and optical properties. When using the OPC for particle sizing, differences in optical properties, influenced by shape and inhomogeneity, between the baseline case and the inversion models result in biases in particle size estimation across different models. Despite the homogeneous super-spheroid models having identical shapes to the baseline case,

differences associated with the inhomogeneity introduce size biases between them. Additionally, for sphere models, deviations in shape from the baseline case, combined with inhomogeneity, further contribute to significant discrepancies in particle size

estimation. Therefore, accurately retrieving *n* is challenging because the scattering coefficients are highly sensitive to the size distribution. However, retrieving *k* from the absorption coefficients is still possible.

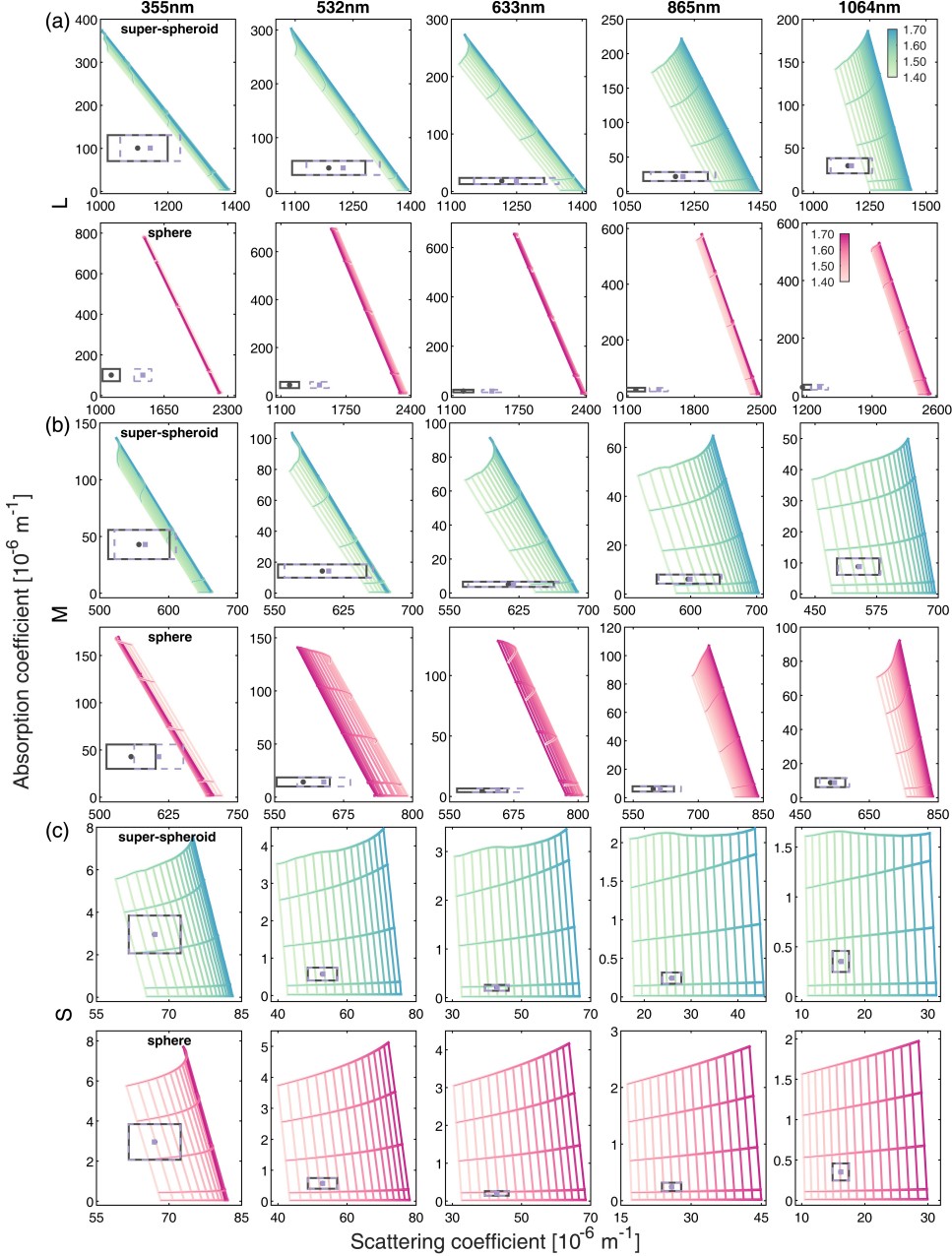

**Figure 8: Results are similar to Figure 5, but represent for E3/E4.**

In Figure 9a, the refractive indices retrieved by various models and methods in E3 are illustrated. Note that the differences in the refractive indices are insignificant between those obtained based on the sphere and super-spheroid models at small sizes. Additionally, in E3 and E4, the refractive indices retrieved from the Bouguer–Lambert method are model-dependent, which can be attributed to inconsistencies in the volume size distribution. The total volume of the dust particle ensemble is substantially smaller for the super-spheroid models compared to the sphere models (Figure 4), resulting in a higher absorptivity of the medium. Consequently, the imaginary parts of the refractive indices are significantly higher for those using the super-spheroid models. Interestingly, the imaginary parts retrieved by the Bouguer–Lambert method using the sphere models (Figure 9a) are closer to the range values for the super-spheroid models (Figure 9b). In Figure 9b, the range values of $k$ retrieved solely from the absorption coefficients for the sphere models are significantly smaller than the counterparts for the super-spheroid models at large sizes. The difference can range from approximately 0.002 in a high-absorption scenario (such as 355 nm) to as low as 0.0007 in a low-absorption scenario (such as 633 nm). However, the difference is insignificant at small sizes due to a relatively small discrepancy in the size distribution.

In comparing the refractive indices retrieved in E1 (Table 3) and E3 (Table 4), it is important to note that the range values of $k$ retrieved using the super-spheroid models are generally consistent in both numerical simulations. However, the values obtained using the sphere models are significantly smaller in E1 compared to E3, particularly at medium and large sizes. The accurate retrieval of the real parts is only possible under very strict conditions in which there are no discrepancies in the size distribution and morphology between the baseline case and the inversion models, as demonstrated in numerical simulation E1. However, it is essential to note that successful retrieval does not guarantee that the inversion model shares identical optical properties with the baseline case (refer to Sect. 3.3). Despite this finding, the exact refractive indices can still be retrieved at small sizes. However, it should be noted that the refractive indices obtained at small sizes may not be applicable to large sizes as they depend on the sizes in the retrieval process when assuming a homogeneous model (Figure 9). The size-dependent refractive indices are more evident in E3/E4 than in E1/E2 due to significant discrepancies in size distribution between the baseline case and the inversion models. Specifically, the imaginary parts decrease as the size increases, as shown in Figure 7 and Figure 9. The variation of imaginary parts with particle size is less evident when the inversion models share an identical size distribution with the baseline case. However, the variation of imaginary parts is more pronounced at 355 nm in E1 compared with other wavelengths, primarily due to the particle's relatively larger size parameter at 355 nm, which leads to variations in optical properties with the size parameter.

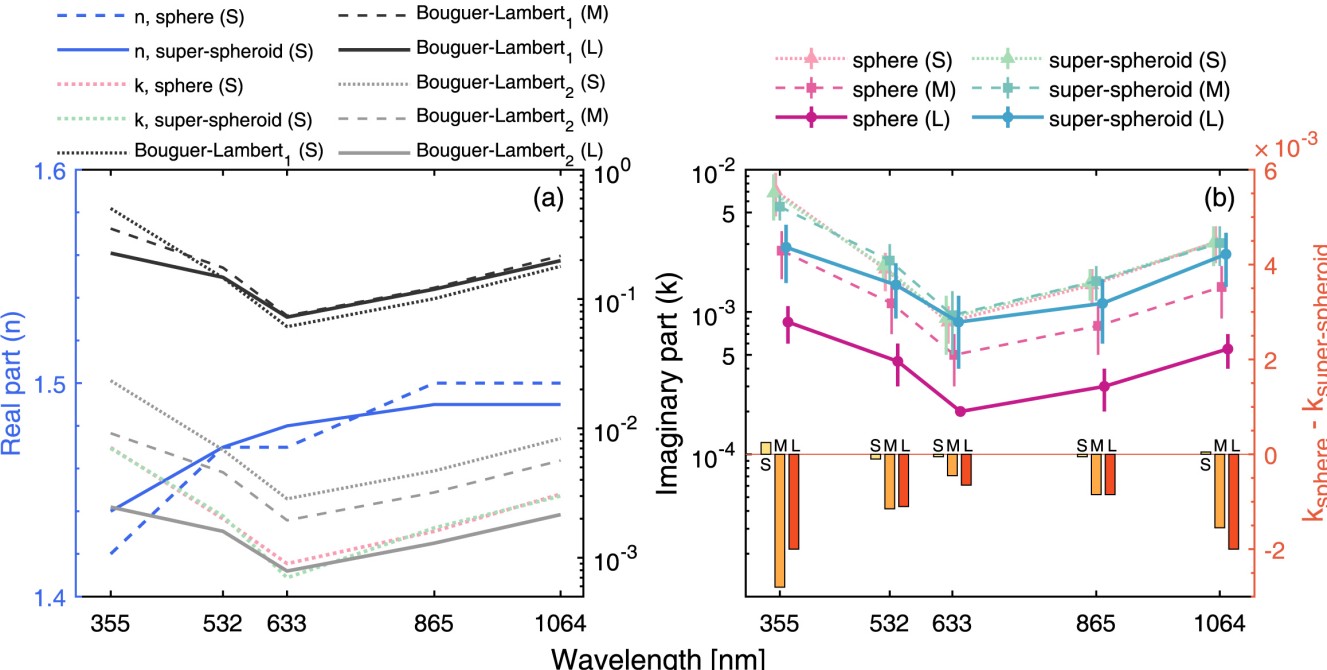

**Figure 9: (a)** The exact refractive indices retrieved from the homogeneous sphere models and the homogeneous super-spheroid models at small sizes (S) and the imaginary parts retrieved by the Bouguer–Lambert method for E3. Subscript 1 indicates that the volume of dust aerosols is calculated from the size distributions for the homogeneous super-spheroid models while the subscript 2 represents volume that for the homogeneous sphere models. **(b)** is similar to Figure 7, but represents for E3.

**Table 4: Results are similar to those in Table 3, but represents for E3.**

| Numerical simulation E3 | | | 355 nm | | 532 nm | | 633 nm | | 865 nm | | 1064nm | |
|---|---|---|---|---|---|---|---|---|---|---|---|---|
| | | | *n* | *k* | *n* | *k* | *n* | *k* | *n* | *k* | *n* | *k* |
| S | sphere | exact | 1.42 | 0.0071 | 1.47 | 0.0020 | 1.47 | 0.0009 | 1.50 | 0.0016 | 1.50 | 0.0031 |
| | | range | 1.40-1.48 | 0.0047-0.0095 | 1.44-1.49 | 0.0014-0.0026 | 1.46-1.50 | 0.0006-0.0011 | 1.48-1.52 | 0.0011-0.0020 | 1.48-1.52 | 0.0022-0.0040 |
| | super-spheroid | exact | 1.44 | 0.0070 | 1.47 | 0.0021 | 1.48 | 0.0007 | 1.49 | 0.0017 | 1.49 | 0.0030 |
| | | range | 1.40-1.52 | 0.0044-0.0093 | 1.45-1.51 | 0.0017-0.0025 | 1.46-1.50 | 0.0005-0.0013 | 1.47-1.51 | 0.0012-0.0020 | 1.47-1.50 | 0.0021-0.0040 |
| M | sphere | exact | nan | nan | nan | nan | nan | nan | nan | nan | nan | nan |
| | | range | nan | 0.0017-0.0037 | nan | 0.0007-0.0016 | nan | 0.0003-0.0007 | nan | 0.0005-0.0011 | nan | 0.0009-0.0021 |
| | super-spheroid | exact | nan | nan | nan | nan | nan | nan | 1.43 | 0.0018 | 1.45 | 0.0028 |
| | | range | nan | 0.0044-0.0066 | nan | 0.0016-0.0030 | nan | 0.0005-0.0014 | 1.40-1.50 | 0.0012-0.0021 | 1.42-1.49 | 0.0021-0.0040 |
| L | sphere | exact | nan | nan | nan | nan | nan | nan | nan | nan | nan | nan |
| | | range | nan | 0.0006-0.0011 | nan | 0.0003-0.0006 | nan | 0.0002-0.0002 | nan | 0.0002-0.0004 | nan | 0.0004-0.0007 |
| | super-spheroid | exact | nan | nan | nan | nan | nan | nan | nan | nan | nan | nan |
| | | range | nan | 0.0016-0.0041 | nan | 0.0009-0.0022 | nan | 0.0004-0.0013 | nan | 0.0006-0.0017 | nan | 0.0015-0.0036 |

## 3.3 Comparison of the optical properties of baseline case with those of inversion models calculated using the retrieved refractive indices

Compared to the fundamental microphysical properties, the variations in the calculated optical properties using different models are of greater concern in practical implementation. We compare the optical properties calculated from different models (sphere and super-spheroid) and different refractive indices, including the extinction coefficients (scattering coefficients + absorption coefficients), single scattering albedo (SSA), and asymmetry factor. The E4 scenario represents a situation closer to real laboratory experiments, while E1 is considered an ideal scenario. In most cases, the discrepancies between the baseline case and the look-up table are so significant that the scattering truncation correction can barely affect the retrieved refractive indices. The differences in optical properties between the results in E1 and E2, in addition to E3 and E4, are negligible. Thus, only the results from E1 and E3 are illustrated in Figure 10. Note that the real parts of the refractive indices cannot be obtained

in most cases in E1 and E3; hence, the real parts are set to 1.52 in such cases based on previous studies (Di Biagio et al., 2019; Dubovik et al., 2002; Müller et al., 2009; Wagner et al., 2012).

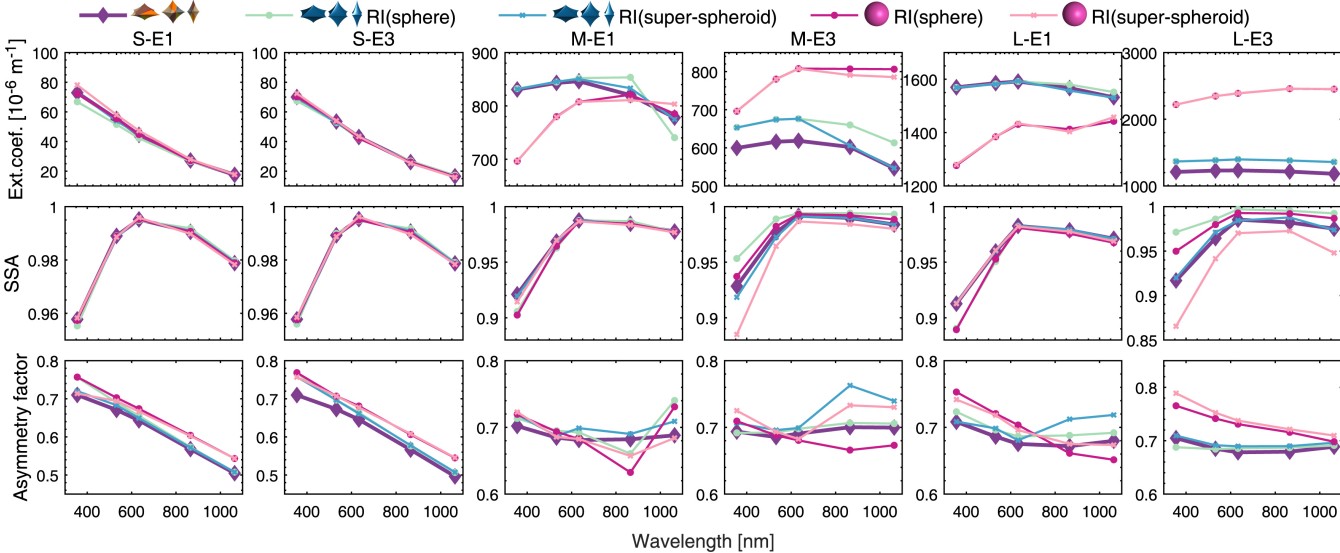

**Figure 10: The optical properties of the baseline case and the inversion models calculated based on different refractive at different sizes for E1 and E3. RI(super-spheroid) indicates that the refractive indices used for optical modelling are the retrieval results from**
**the homogeneous super-spheroid models while RI(sphere) means that those are from the homogeneous sphere models.**

In E1, the size distributions are the same among various models, and no significant differences in the retrieved refractive indices between the super-spheroid and sphere models are found. However, when using the homogeneous super-spheroid models, the calculated optical properties are generally closer to those of the baseline case compared to those using the sphere models in E1, emphasizing the importance of the model shape in simulating nonspherical dust aerosols. In E3, the differences
in optical properties between the super-spheroid and sphere models are further amplified by the discrepancies in size distributions. These differences become more significant as the size increases.

The SSA is highly sensitive to the imaginary parts of the refractive indices. The SSA calculated using the super-spheroid models with RI(super-spheroid) show good agreement with the baseline case in E3. However, when using the sphere models or the super-spheroid models with RI(sphere), the results vary significantly from the baseline case in M-E3 and L-E3.
In particular, the SSA for L-E3 shown in Figure 10 suggests that the imaginary parts retrieved from sphere models are underestimated. By utilizing the sphere model, the SSA calculated using the refractive indices retrieved from the sphere model is found to be larger than that of the baseline case, particularly at larger sizes, by as much as 0.03, especially under conditions of high absorption at 355 nm. The underestimation in the imaginary parts of the sphere models results from significant discrepancies in the size distributions between the sphere model and the baseline case. When assuming a spherical model, the
OPC provides larger sizes than the baseline case, which increases absorption coefficients of the model on a larger scale. As a result, the retrieved imaginary parts decrease.

The asymmetry factor is found to be more sensitive to the shape of models than the SSA, especially at large sizes. This finding is consistent with previous studies (Mishchenko et al., 1996; Otto et al., 2009). Generally, the sphere model exhibits a significant decreasing trend with wavelength and tends to overestimate the asymmetry factor, particularly at relatively short

wavelengths. For instance, the overestimation can reach up to 0.05 at 355 nm for L-E3. However, this decreasing trend is not evident at large sizes for the baseline case. The discrepancy in the asymmetry factor may introduce a significant bias in climate modelling. Despite notable advancements, many climate models still rely on sphere models to simulate dust aerosols (Balkanski et al., 2007; Danabasoglu et al., 2020; Hess et al., 1998; Hurrell et al., 2013; Liu et al., 2016; Mishchenko et al., 1995).

Significant variations are observed in the asymmetry factor at wavelengths of 865 nm and 1064 nm for M-E1 and M-E3 (Figure 10). These variations are attributed to the variations in the real parts of the refractive indices. To simplify the discussion, the refractive indices retrieved from the homogeneous super-spheroid models are referred to as RI(super-spheroid), while those from the homogeneous sphere models are referred to as RI(sphere). Below 865 nm, the real parts of RI(sphere) are set to the default value of 1.52 for M-E1, and the same is done for RI(super-spheroid) for M-E3 as the target values deviate significantly

from the values in the look-up table. However, at 865 nm and 1064 nm, the target values fall within the look-up table, and the extinction coefficients are well matched in M-E1 and M-E3. Despite this, the retrieved real parts deviate significantly from the value of 1.52. Interestingly, the results imply that fixing the real parts to a value of 1.52 for all five selected wavelengths will be a better choice than using the retrieved values to reproduce the asymmetry factor of the baseline case. Hence, it is reasonable and essential to choose a representative value for the real parts, which is necessary in the retrieval because of the discrepancies

in the scattering coefficients of the inversion models.

The significant variations in the asymmetry factor also indicate that, despite the good agreement in the scattering and absorption coefficients between the baseline case and the inversion models, it does not guarantee accurate simulation of all the optical properties. For instance, reproducing the asymmetry factor calculated from the inhomogeneous models is challenging. This difficulty implies an inherent defect in homogeneous models, a finding that is also consistent with previous studies (Zong et

al., 2021).

In Figure 11, the phase matrices of the baseline case and those calculated based on the sphere and super-spheroid models are illustrated. Note that the imaginary parts are large at small sizes, whereas they are small at large sizes. The trends in the phase matrices are mainly determined by the morphology of the particles. Significant discrepancies are observed between the results from the homogeneous sphere models and the target values. However, the results from the homogeneous super-spheroid

models are in good agreement with the target values, especially in the low absorption scenario (i.e., L-E1 and L-E3). Nonetheless, the phase function ($P_{11}$) is less sensitive to the particle shape in the scenario of high absorption (i.e., S-E1 and S-E3). Additionally, notable differences in the $-P_{12}/P_{11}$ and $P_{22}/P_{11}$ are found between those calculated by the super-spheroid models using RI(super-spheroid) and RI(sphere) for S-E1. These differences can be attributed to the variations in the real parts of the refractive indices. The optical properties in Figure 10 may imply that the differences between the homogeneous and

inhomogeneous super-spheroid models are negligible at small sizes. However, Figure 11 indicates that significant

discrepancies in the phase matrices still exist at small sizes, particularly in the backward direction. Nevertheless, for large sizes, the uncertainties in the phase matrices resulting from different refractive indices obtained using various models are deemed acceptable, both for the super-spheroid model and the sphere model.

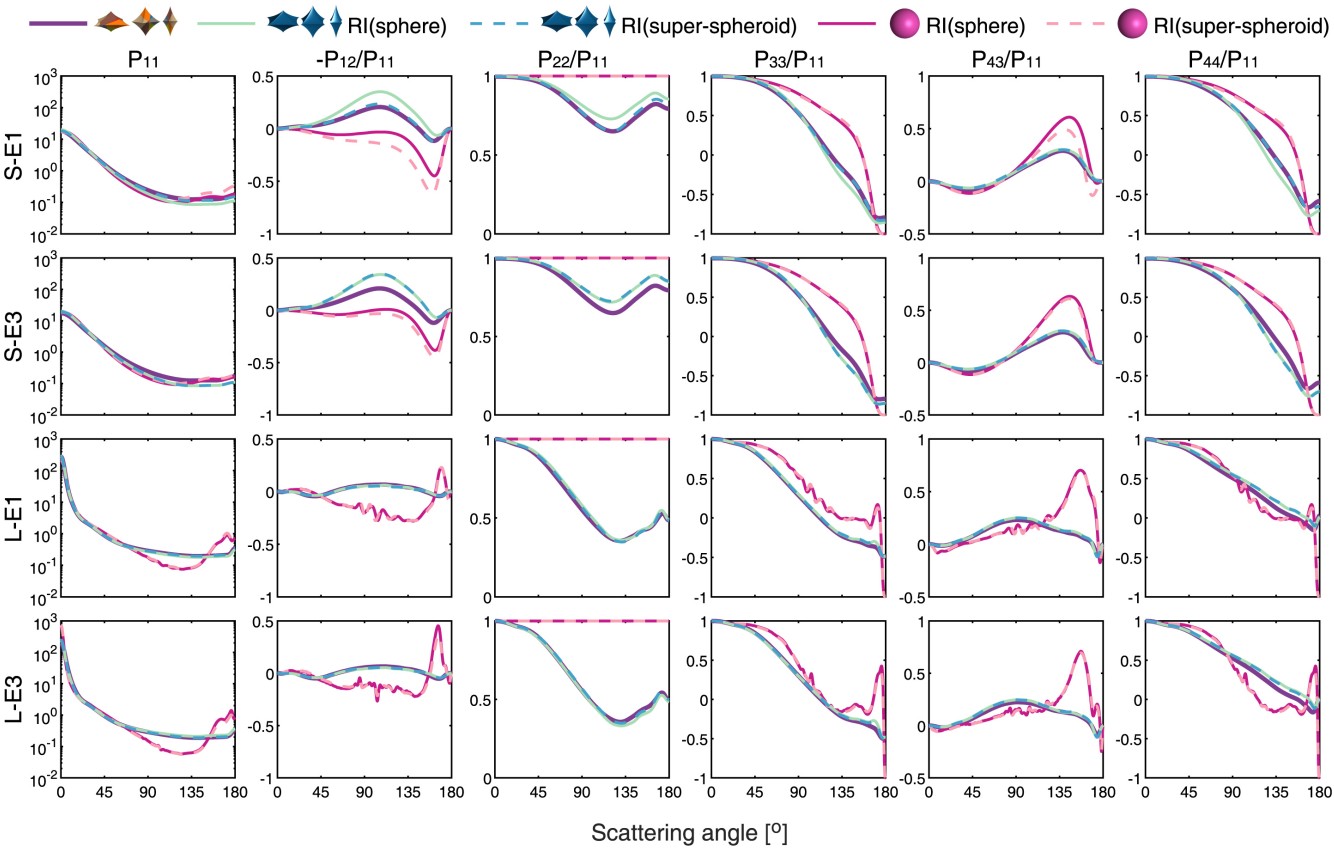

**Figure 11: The phase matrices of the baseline case and the inversion models calculated based on different refractive indices at a wavelength of 355 nm and different sizes (S and L) for E1 and E3. RI(super-spheroid) indicates that the refractive indices used for optical modelling are the retrieval results from the homogeneous super-spheroid models, while RI(sphere) means that those are from the homogeneous sphere models.**

### 3.4 Discussion about the actual laboratory scenario

It is not scientifically reasonable to quantitatively compare the refractive indices and the optical properties obtained in this study with those from actual laboratory measurements due to several assumptions made in the numerical simulations. For instance, we assume that the refractive indices of various minerals are accurate. However, significant uncertainties in the refractive indices of hematite can be noted, and the refractive indices of goethite are only available for limited wavelengths and are based on a single study (Go et al., 2022). Besides, instrumental error is more complicated in the real world and discrepancies in size distribution between the model and the realistic particles exist. Nevertheless, the results in the numerical simulations provide a reference for investigating the extent to which the uncertainties resulting from the assumption of spherical particles can affect the actual laboratory measurements.

Previous studies have proven that the super-spheroid models with a roundness parameter ($e$) value of 2.5 exhibit comparable optical properties to realistic dust aerosols (Kong et al., 2022; Lin et al., 2018, 2021). To quantify the non-sphericity of irregular particles, two metrics, the shape index ($SI$; Sun et al., 2021) and the degree of sphericity (Saito and Yang, 2022) have been proposed. The shape index ($SI$) is defined as follows:

$$SI = \frac{3V}{4\pi\left(A_{\text{proj}}/\pi\right)^{3/2}}, \tag{8}$$

where $V$ denotes the volume of a particle and $A_{\text{proj}}$ indicates the mean projected area of a particle. The degree of sphericity ($\Psi$) shares the essence of $SI$ and can be expressed as:

$$\Psi = \frac{\pi^{1/3}(6V)^{2/3}}{A_{\text{proj}}}. \tag{9}$$

It is evident that the conversion between $SI$ and $\Psi$ can be obtained using the following formula:

$$SI = \Psi^{3/2}. \tag{10}$$

The degree of sphericity ($\Psi$) for actual dust aerosols was found to range from 0.58 to 0.79, as calculated from the morphological measurements (Saito and Yang, 2022). Correspondingly, the values of shape index ($SI$) for actual dust aerosols range from 0.43 to 0.70. Figure 12 illustrates the shape index for various models (sphere case, spheroid case, D06 case and super-spheroid case). The $SI$ equals 1 for a sphere, and it decreases as non-sphericity increases. While spheroid models with a broad range of aspect ratios (e.g., from 0.3 to 3.0) are useful for fitting measurements effectively (Dubovik et al., 2006; D06 case), we only consider three aspect ratios (0.5, 1.0 and 2.0), which are consistent with the target shapes. The $SI$ of the spheroid case is slightly smaller than 1, whereas the D06 case has a much smaller $SI$ value. It is worth noting that the super-spheroid case, which share an identical shape with the baseline case, exhibit comparable non-sphericity to actual dust aerosols.

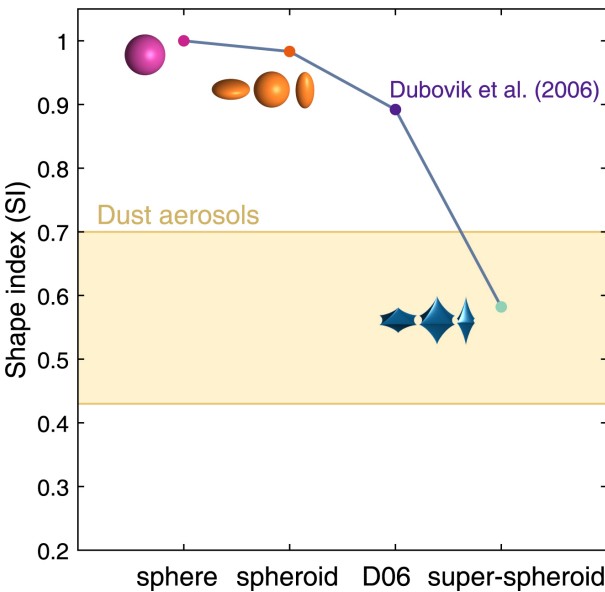

**Figure 12. The shape index (*SI*) of various models (sphere case, spheroid case, D06 case and super-spheroid case). The yellow area indicates the range of the shape index of the actual dust aerosols. The D06 case indicates the spheroid models developed in Dubovik et al. (2006).**

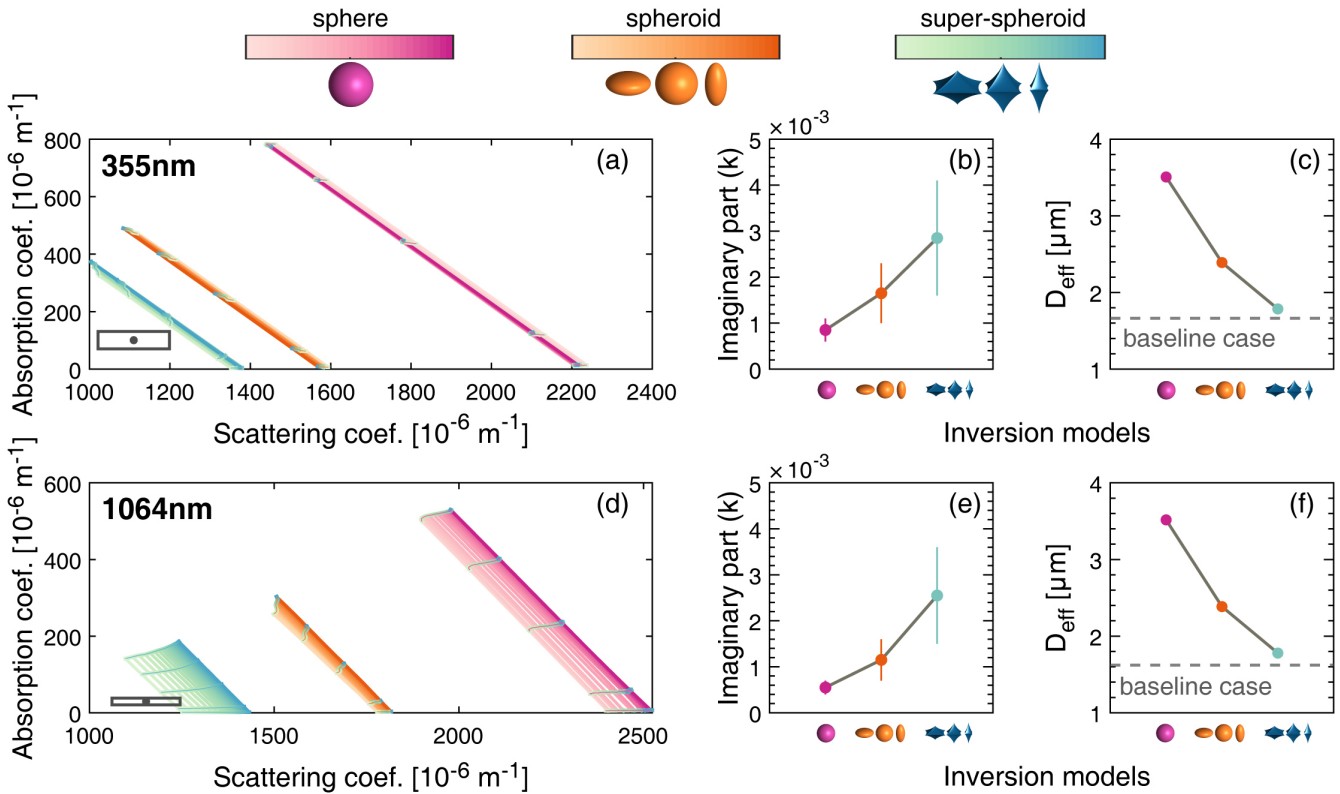

**Figure 13. (a, d)** The target values of baseline case and the look-up tables for refractive indices produced by three models (sphere case, spheroid case and super-spheroid case) with the size distribution (L) at 355 nm (a) and 1064 nm (d) for E3. The refractive indices of the look-up tables are consistent with those presented in Figure 5. **(b, e)** The imaginary parts of the refractive indices retrieved from the absorption coefficients by the three models at 355 nm (b) and 1064 nm (e), respectively. The error bar denotes half of the range of the retrieved imaginary parts, and the marker indicates the mean of the range. **(c, f)** The effective diameter ($D_{eff}$) calculated by various models at 355 nm (c) and 1064 nm (f), respectively. The dashed line denotes the effective diameter of the baseline case.

Considering potential underestimations of the non-sphericity of actual dust aerosols by inversion models, we utilize 3 models (sphere case, spheroid case and super-spheroid case) to depict various retrieval scenarios. The retrieval process is consistent with the discussions presented in Sect. 3.1 and 3.2. Note that the sphere case and the spheroid case indicate inversion models with larger *SI* values compared to the baseline case (less irregular), while the super-spheroid case represents inversion models sharing identical shapes with the baseline case.

Figure 13 illustrates the retrieval results obtained from different inversion models in L-E3 at 355 nm and 1064 nm. While discrepancies exist between the look-up tables and the target values, the inversion models generally yield values closer to the target values as non-sphericity approaches that of the baseline case (Figure 13a, d).

We employ the effective diameter ($D_{eff}$) to characterize the size for an ensemble of particles. The $D_{eff}$ is defined as follows(Hansen, 1971):

$$D_{\text{eff}} = \frac{\int D_{\text{geo}}{}^3 \frac{\mathrm{d}N}{\mathrm{d}D_{\text{geo}}} \mathrm{d}D_{\text{geo}}}{\int D_{\text{geo}}{}^2 \frac{\mathrm{d}N}{\mathrm{d}D_{\text{geo}}} \mathrm{d}D_{\text{geo}}}, \qquad (11)$$

where $D_{\text{geo}}$ indicates the geometric diameter and $\frac{\mathrm{d}N}{\mathrm{d}D_{\text{geo}}}$ denotes the size distribution. Note that the $D_{\text{eff}}$ of the sphere model is

significantly larger than the baseline case (Figure 13c, f), which contributes the evident discrepancy between the look-up tables and the target values.

Generally, the $D_{\text{eff}}$ of inversion models converges toward that of the baseline case as non-sphericity approaches that of the baseline case (Figure 13c, f). It is worth noting that the retrieved imaginary parts of the refractive indices and their uncertainties increase as $D_{\text{eff}}$ decreases (Figure 13b, c, e, f). This trend arises from the decrease in scattering coefficients and absorption

coefficients as $D_{\text{eff}}$ decreases, leading to a contraction in the overall shape of look-up tables. However, the target values and their associated uncertainties remain constant, resulting in the uncertainties of the target values encompassing a broader range of the look-up tables. The differences between the sphere case and super-spheroid case are approximately 0.002 at both 355 nm and 1064 nm, which is notably large compared to the retrieved values by the sphere model (0.00085 at 355 nm and 0.00055 at 1064 nm). The uncertainties resulting from the assumption of sphere models for dust aerosols may be substantial in actual

laboratory experiments. Note that the optical properties generated by the super-spheroid models with RI(super-spheroid) remain largely consistent with the baseline case, as shown in Figure 10 and Figure 11. Therefore, exploring nonspherical models for the retrieval, especially those exhibiting comparable non-sphericity to the actual dust aerosols, could be beneficial. However, retrieval for the real parts remains challenging in retrieval from the scattering coefficients and absorption coefficients, primarily due to the high sensitivity of scattering coefficients to particle size and shape. Thus, obtaining the real

parts may require alternative methods, additional measurements of other metrics (e.g., phase function), or simply setting them to a representative value (Dubovik et al., 2006; Grams et al., 1974; Patterson et al., 1977; Sklute et al., 2015; Sokolik et al., 1993).

It is noteworthy that the uncertainties in refractive indices may be further amplified by the presence of exceptionally large particles in the atmosphere (Adebiyi et al., 2023a), which are not considered in this study. Such concerns arise due to the fact

that biases in extinction coefficients (scattering coefficients + absorption coefficients) increase with discrepancies in the size distribution. Given the size-dependent nature of the obtained refractive indices, it is crucial to account for the size distribution of the sample when employing laboratory-derived refractive indices for simulation. It is advantageous to prepare samples of varying sizes for the refractive index retrieval in a laboratory setting, thereby offering size-dependent refractive indices. The laboratory equipment generally allows for control over the maximum size of dust samples, providing some level of control

over this factor.

## 4 Conclusions

Dust aerosols are rarely homogeneous and spherical. However, when determining their refractive indices in the laboratory, it is often assumed that the particles are homogeneous and spherical. These refractive indices are then used for optical calculations for either spheres or nonspherical particles with downstream applications. In this study, we conducted a theoretical investigation to explore the uncertainties associated with this rationale for laboratory measurements of refractive indices for dust aerosols in the wavelength range of 355 to 1064 nm. Additionally, we aimed to determine the impact of these uncertainties on the optical properties of dust aerosols. This is a crucial step in validating the fundamental microphysical properties of dust aerosols and understanding the extent of uncertainties before applying them in specific research.

The uncertainties in the refractive indices arise from the imperfect representation of the non-sphericity and inhomogeneity of realistic dust aerosols by the inversion models. This imperfect representation leads to discrepancies in the size distribution when using the OPC for particle sizing, which in turn results in discrepancies in the scattering and absorption coefficients. Consequently, uncertainties arise when retrieving refractive indices from scattering and absorption coefficients. Moreover, the retrieved refractive indices are found to be size-dependent primarily due to discrepancies in size distribution. As the size increases, the imaginary parts decrease.

Accurately retrieving the refractive indices, particularly the real parts, is challenging due to the high sensitivity of scattering coefficients to particle size and shape. It is recommended to choose a representative value for the real parts in the retrieval process. However, it remains possible to retrieve the imaginary parts solely from the absorption coefficients. The optical properties derived from the super-spheroid models, utilizing the imaginary parts retrieved by the super-spheroid models, are generally consistent with the baseline case. Therefore, nonspherical models, especially those reflecting similar non-sphericity to actual dust aerosols, are highly recommended in the retrieval process for dust aerosols, particularly at short wavelengths.

The sphere model tends to underestimate the imaginary parts of the refractive indices, correspondingly overestimate the SSA, while the super-spheroid models are generally in good agreement with the baseline case. The imaginary parts retrieved for the sphere model are approximately 0.0009 and 0.0002 at 355 nm (high absorption conditions) and 633 nm (weak-absorption conditions), respectively. However, the differences between the sphere model and the super-spheroid models can reach up to approximately 0.002 under high absorption conditions and 0.0007 under weak-absorption conditions. While the asymmetry factor and the phase matrix are more sensitive to particle shape, the differences in the imaginary parts have minimal impact on them. It is essential for accurate dust simulation to utilize the nonspherical models and the refractive indices retrieved from these models.

The findings of this study provide valuable insights into the uncertainties associated with currently available laboratory-measured refractive indices (Di Biagio et al., 2019). It is important to note that the optical properties of inhomogeneous particles cannot be fully characterized based on homogeneous models. However, there is still a long way to go in developing a comprehensive and suitable database of inhomogeneous dust models that can be applied in various fields such as remote sensing and climate models (Wang et al., 2022). Previously, homogeneous models were the only option, but the ultimate goal

is to accurately characterize realistic dust aerosols using inhomogeneous models. Further efforts to improve the computational

efficiency are crucial for calculating the optical properties of nonspherical particles with large size parameters. The use of GPU-accelerated computing and data-driven techniques shows promise in this regard (Bi et al., 2022; Yu et al., 2022). Additionally, future work should focus on studying the refractive indices of individual minerals, as well as the mineral composition and internal structure of dust aerosols, rather than solely focusing on the refractive indices of entire dust particles.

## Data availability

The data used in the numerical simulations are available at https://zenodo.org/records/11093920. The additional data from this study are available on request.

## Author contribution

SK and LB designed the numerical experiments and SK carried them out. ZW developed the code for calculating the optical properties of inhomogeneous models. SK prepared the manuscript with contributions from all co-authors. LB revised the manuscript.

## Competing interests

The authors declare that they have no conflict of interest.

## Acknowledgements

This work was funded by National Natural Science Foundation of China (42022038, 42090030). We would like to thank the two anonymous reviewers for their invaluable help in improving the manuscript. We acknowledge the support of computation by the cluster at State Key Lab of CAD&CG at Zhejiang University, the computing facilities at China HPC Cloud Computing Center, and the National Key Scientific and Technological Infrastructure project "Earth System Numerical Simulation Facility" (EarthLab).

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
