# Peer review of "The uncertainties in the laboratory-measured short-wave refractive indices of mineral dust aerosols and the derived optical properties: A theoretical assessment"

_EGUsphere, 2023_

## Referee Comment (RC1)

**Review of "The uncertainties in the laboratory-measured short-wave refractive indices of mineral dust aerosols and the derived optical properties: A theoretical assessment" by Senyi Kong et al.**

This study investigates differences in the retrieval refractive complex refractive indices using different optical models. In their baseline case, optical properties were calculated assuming inhomogeneous super-spheroid model, and were then compared with homogeneous super-spheroid models and the homogeneous sphere models. Optical properties were calculated for wavelengths 355, 532, 633, 865, 1064 nm. Three particle size distributions (S, M, and L) or combinations of them were used in the simulations.

Overall, the work is interesting, and the evaluation of different aerosol optical models is relevant to the scientific community. However, certain sections of the text are not clear or are confusing. For instance, the authors refer to a numerical simulation as a "measurement" and, at times, as "experiments". Additionally, the baseline simulation is referred to as a "dust sample". These nomenclature makes some parts of the text unclear. Also, the authors evaluate the retrieval refractive indices in what they called "ideal" and "realistic" scenarios. However, the "ideal case" corresponds to a simulation where size and scattering truncation are not corrected, and it is unclear why they would evaluate and label such a case as "ideal".

In the "realistic case", which appears to be the appropriate scenario for evaluation, they found that using either the homogeneous super-spheroid model or the homogeneous sphere model was unable to match the optical properties for the large particle sizes. The reason for this is attributed to "differences in size distribution between the dust samples and the models." However, it is not clear why there are differences in the size distribution between the baseline and the model simulations.

**Specific Comments:**

Title: The title mentions short-wave refractive indices only, but the manuscript evaluates five wavelengths 355, 532, 633, 865, and 1065nm, ranging from near ultraviolet (NUV) to near infrared (NIR).

Page 2, Line 36: It would be useful to include a brief description of what a homogeneous super-spheroid model actually is.

Page 4, Line 99: "*Dust samples of different sizes and optical databases for homogeneous models were prepared. Inhomogeneous super-spheroid dust models were regarded as the "dust samples."*". It is not clear what the authors mean by "prepared". Do you mean selected? From where? Which datasets were used to define the "dust samples"? It would be more clear if you define "dust samples" as the "baseline case".

Page 4, Line 99-108: For clarity, I would suggest the authors to rewrite the steps of the numerical experiment in a more directly, avoiding stating that quantities were measured, and refrain from calling an optical model/simulation a "dust sample".

Page 4, Line 113: "*Two correction processes were considered for the measurements of "dust samples" to mimic actual laboratory experiments*." This sentence is not clear. Were these processes used in the numerical simulation? If so, please make sure to clarify that in the sentence.

Page 4, Line 123: For clarity, I suggest replacing: "experiments…" by "numerical simulations", here and throughout the entire manuscript.

Page 4, Line 124: "*Note that E1 represents an ideal situation in which no instrument defects need to be corrected for the measurements*, …" It is not clear what the authors are calling "instrument defects" here. The corrections the authors mentioned previously (scattering truncation and size) corrections should not be refer to as instrument defects. The corrections are needed because instruments are limited in their measurement capacity and not because of a defect (broken or failure).

Table 1, Caption: Does the instrument bias the authors are referring to correspond to the same two corrections mentioned previously?

Page 11, Line 275: It is not clear why the authors decided not to include uncertainties in the absorption coefficient measurements in their simulations. The authors mentioned "*corrections were validated using measurements*". Why does the validation of corrections by measurements change the necessity of including them in the simulations?

Page 12, Line 285: "*Due to the sparce nature of the measurements, it was almost impossible for them to fall on the grid points of the look-up tables… These average refractive indices were referred to as the exact values*." This sentence requires clarification. What measurements are you referring to? Are the authors saying that the calculated scattering and absorption coefficients could not be obtained with the optical parameters represented in the authors' look-up tables?

Page 17, Line 384: "*Note the significant discrepancy emerged between the "dust samples" and the homogeneous super-spheroid models as the size increased, which was inconsistent with the findings in Figure 5*." Can the authors explain why the discrepancy is larger in this case (E3/E4) compared to E1/E2? Specifically, in the E4 case where the model has both scattering and size correction, why would the discrepancy be larger in this case?

Page 21, Line 428: "*The E4 scenario represented measurements closer to those obtained in the laboratory, while E1 was considered an ideal scenario*." According to Table 1, E4 corresponds to the cases where both size and scattering truncation corrections were applied, while E1 corresponds to a case "without correction". It is not clear why the authors would call the case without correction "ideal".

Page 22, Line 441: Instead of using the term "measurement", please clarify what exactly you are referring to. The same applies to lines 446, 447, and 450.

Figure 11: It is difficult to distinguish between each line in the plots. Using different colors and/or line styles may improve clarity.

Page 25, Line 518: "*However, when measuring their refractive indices in the laboratory, it is ...*". Refractive indices of aerosol particles are not usually directly measured in the laboratory, but are derived from absorption or extinction measurements assuming an optical model.

Page 25, Line 530: "*Under and ideal scenario, where no instrumental defects needed to be corrected, the look-up tables for the homogeneous super-spheroid models were able to fit the measurements at any size.*". There are a few issues here:

First, the size and scattering truncation corrections are not "defects". A nephelometer measuring scattering in an angular range from 7 to 170 degrees is not a defect. Extrapolation (or correction) is the procedure used to obtain the total integrated scattering. Second, I would not refer to an "ideal scenario" as a case where beta_scat(0 to180) is approximated by beta_scat(7 to 170). Was a solution found using the super-spheroid model fitted with an approximation beta_scat(7 to 170)? What does that say about the consistency of the model?

Page 26, Line 552: "*The retrieved refractive indices were found to be size-dependent. As the size increased, the imaginary part decreased.*". Where is this shown in the authors analysis? How do you explain this result if you have assumed the same composition in each simulation? Was that observed for both the homogeneous and inhomogeneous models?

Page 26, Line 556: Please define "true values". Are the authors referring to the complex refractive indices obtained with the inhomogeneous super-sphere model? Why are they not shown?

---

## Author Comment (AC1)

In this response letter, comments from the reviewer are highlighted in black font, our responses are in blue font, and modifications made to the manuscript are indicated in red font.

**Reviewer #1**

This study investigates differences in the retrieval refractive complex refractive indices using different optical models. In their baseline case, optical properties were calculated assuming inhomogeneous super-spheroid model, and were then compared with homogeneous super-spheroid models and the homogeneous sphere models. Optical properties were calculated for wavelengths 355, 532, 633, 865, 1064 nm. Three particle size distributions (S, M, and L) or combinations of them were used in the simulations.

Overall, the work is interesting, and the evaluation of different aerosol optical models is relevant to the scientific community. However, certain sections of the text are not clear or are confusing. For instance, the authors refer to a numerical simulation as a "measurement" and, at times, as "experiments". Additionally, the baseline simulation is referred to as a "dust sample". These nomenclature makes some parts of the text unclear. Also, the authors evaluate the retrieval refractive indices in what they called "ideal" and "realistic" scenarios. However, the "ideal case" corresponds to a simulation where size and scattering truncation are not corrected, and it is unclear why they would evaluate and label such a case as "ideal".

In the "realistic case", which appears to be the appropriate scenario for evaluation, they found that using either the homogeneous super-spheroid model or the homogeneous sphere model was unable to match the optical properties for the large particle sizes. The reason for this is attributed to "differences in size distribution between the dust samples and the models." However, it is not clear why there are differences in the size distribution between the baseline and the model simulations.

Thank you for providing us with detailed and valuable feedback. We greatly appreciate your insights and concerns regarding certain sections of the text.

1. Nomenclature Clarification:

We acknowledge the confusion might be caused by referring to numerical simulations as 'measurements' or 'experiments' and the baseline simulation as a 'dust sample'. To accurately reflect the nature of our simulations, we have revised the terminology by using terms 'numerical simulation' and 'baseline case' to avoid potential ambiguity.

2. Clarification of 'Ideal' Case:

We have revised the explanation regarding the 'ideal' case in the text to emphasize that it refers to a scenario where the size distributions and scattering coefficients of the baseline case can be accurately obtained and used in the retrieval. This implies that the inversion models share an identical size distribution with the baseline case. Additionally, we have made corresponding revisions to Table 1 in the manuscript to avoid any confusion regarding the terminology used.

3. Explanation of Differences in Size Distribution:

Essentially, the inversion models imperfectly represent the non-sphericity and inhomogeneity of the baseline case. This imperfect representation leads to discrepancies in the size distribution when using the OPC for particle sizing. We have provided a more detailed

explanation of this difference in relevant sections of the main text. Additionally, we have included a more comprehensive analysis of the differences in size distribution between the baseline and model simulations in Sect. 3.4.

**Specific Comments:**

Title: The title mentions short-wave refractive indices only, but the manuscript evaluates five wavelengths 355, 532, 633, 865, and 1065nm, ranging from near ultraviolet (NUV) to near infrared (NIR).

We understood that this term "short-wave" may have different wavelength ranges in different contexts. For example, Di Biagio et al. (2019) use the term to refer to the 370–950 nm range, while Go et al. (2022) define it as 0.185–4.0 μm to distinguish the terrestrial radiation and the solar radiation. In our study, we specifically focus on the wavelength range of 355 to 1064 nm, as stated in the abstract.

To address this concern, we have added an explanation in the introduction section when the term "short-wave" is first introduced.

"It should be noted that the short-wave (355-1064 nm in this study) refractive indices of dust…"

Page 2, Line 36: It would be useful to include a brief description of what a homogeneous super-spheroid model actually is.

Thank you for your suggestion. We have included a brief introduction to the super-spheroid model in this section.

"To improve upon the spheroid model, the super-spheroid model, which extends the dimensions of both the sphere and spheroid models, has been proposed to provide a more comprehensive framework for describing the shape of dust particles. Initial studies have shown that homogeneous super-spheroid dust models align well with laboratory measurements…"

Page 4, Line 99: "Dust samples of different sizes and optical databases for homogeneous models were prepared. Inhomogeneous super-spheroid dust models were regarded as the "dust samples."". It is not clear what the authors mean by "prepared". Do you mean selected? From where? Which datasets were used to define the "dust samples"? It would be more clear if you define "dust samples" as the "baseline case".

Thank you for your feedback. We have revised these sentences and used the term of "baseline case" instead of "dust samples".

Page 4, Line 99-108: For clarity, I would suggest the authors to rewrite the steps of the numerical experiment in a more directly, avoiding stating that quantities were measured, and refrain from calling an optical model/simulation a "dust sample".

Thank you for your suggestion. We have revised the description of the numerical simulations to avoid ambiguity.

Page 4, Line 113: "Two correction processes were considered for the measurements of "dust

samples" to mimic actual laboratory experiments." This sentence is not clear. Were these processes used in the numerical simulation? If so, please make sure to clarify that in the sentence.

Thank you for your suggestion. We have made revisions to clarify the correction processes used in the numerical simulation.

"In the numerical simulations, we have incorporated two correction processes based on the actual laboratory experiments. The first correction is the size correction, which is employed to determine the geometric size of the particles from imaginary OPC measurements…The second correction is the scattering truncation correction, which is associated with the unavoidable technical limitations in measurements of scattering coefficients."

Page 4, Line 123: For clarity, I suggest replacing: "experiments…" by "numerical simulations", here and throughout the entire manuscript.

Thank you for your suggestion. We have made the corrections accordingly.

Page 4, Line 124: "Note that E1 represents an ideal situation in which no instrument defects need to be corrected for the measurements, …" It is not clear what the authors are calling "instrument defects" here. The corrections the authors mentioned previously (scattering truncation and size) corrections should not be refer to as instrument defects. The corrections are needed because instruments are limited in their measurement capacity and not because of a defect (broken or failure).

Thank you for your feedback. You are correct. The term 'instrument defects' cannot accurately represent the corrections needed for measurements. We have revised these sentences and referred to the 'instrument defects' as unavoidable technical limitations.

Table 1, Caption: Does the instrument bias the authors are referring to correspond to the same two corrections mentioned previously?

Thank you for your feedback. We have revised Table 1 and its caption in the manuscript to address the confusion. The updated table is as follows:

**Table 1: A brief description of the four numerical simulations. Target scattering coefficient denotes the scattering coefficient of the baseline case.**

| Numerical simulations | | Target scattering coefficient used for inversion models | |
| --- | --- | --- | --- |
| | | the same as the baseline case | the scattering coefficients of the baseline case with scattering truncation correction |
| Particle size of inversion models | the same size as the baseline case | E1 | E2 |
| | the size of the baseline case with size correction | E3 | E4 |

Page 11, Line 275: It is not clear why the authors decided not to include uncertainties in the absorption coefficient measurements in their simulations. The authors mentioned "corrections were validated using measurements". Why does the validation of corrections by measurement change the necessity of including them in the simulations?

Thank you for your comment. The corrections for absorption coefficients are conducted using empirical formulas in the laboratory measurements. It is difficult to verify these complex corrections through numerical simulations. These corrections were validated in (Di Biagio et al. (2017a). Therefore, we assume that the absorption coefficient can be obtained accurately within a certain uncertainty, which is set to 30% based on previous studies (Di Biagio et al., 2019).

We have revised the sentence to clarify the process of correcting dust particles and the assumption made regarding the absorption coefficient in the numerical simulations. The revised sentence is as follows:

"Given that the corrections for dust particles are conducted using empirical formulas and are challenging to verify through numerical simulations. In addition, the corrections have been validated using measurements from the Multi-Angle Absorption Photometer (MAAP) and Cavity Attenuated Phase Shift Extinction (CAPS) (Di Biagio et al., 2017a). As a result, we assume that the absorption coefficient can be accurately obtained within a certain uncertainty, and thus, no additional correction is applied in the numerical simulations."

Page 12, Line 285: "Due to the sparce nature of the measurements, it was almost impossible for them to fall on the grid points of the look-up tables… These average refractive indices were referred to as the exact values." This sentence requires clarification. What measurements are you referring to? Are the authors saying that the calculated scattering and absorption coefficients could not be obtained with the optical parameters represented in the authors' look-up tables?

Apologies for any confusion. What we meant is that it is unlikely for the target value (the scattering coefficients and absorption coefficients of the baseline case) to perfectly align with the grid points of the look-up tables due to their sparse nature. Therefore, we took the average of the four refractive indices corresponding to the four smallest values of RMSD and used that average as the exact retrieved value. The revised sentence is as follows:

"However, due to the sparse nature of the look-up table, it is highly unlikely for the target value to fall directly on the grid points. Therefore, we average the four refractive indices corresponding to the four smallest values of RMSD."

Page 17, Line 384: "Note the significant discrepancy emerged between the "dust samples" and the homogeneous super-spheroid models as the size increased, which was inconsistent with the findings in Figure 5." Can the authors explain why the discrepancy is larger in this case (E3/E4) compared to E1/E2? Specifically, in the E4 case where the model has both scattering and size correction, why would the discrepancy be larger in this case?

Apologies for the confusion about the explanation of E1/E2. E1 refers to a scenario where the size distributions and scattering coefficients of the baseline case can be accurately obtained and used in the retrieval. This implies that the inversion models share an identical size distribution with the baseline case. However, in E3/E4, the size distributions of the inversion

models are derived from the size distribution of the baseline case with size correction. We have revised the explanation of E1/E2 accordingly.

Page 21, Line 428: "The E4 scenario represented measurements closer to those obtained in the laboratory, while E1 was considered an ideal scenario." According to Table 1, E4 corresponds to the cases where both size and scattering truncation corrections were applied, while E1 corresponds to a case "without correction". It is not clear why the authors would call the case without correction "ideal".

Apologies for the confusion. The previous reply clarifies that E1 refers to a scenario where the size distributions and scattering coefficients of the baseline case can be accurately obtained and used in the retrieval.

Page 22, Line 441: Instead of using the term "measurement", please clarify what exactly you are referring to. The same applies to lines 446, 447, and 450.

Thank you for your suggestion. We have replaced the term "measurement" with "baseline case" to indicate the optical properties of the baseline case directly, thereby avoiding ambiguity.

Figure 11: It is difficult to distinguish between each line in the plots. Using different colors and/or line styles may improve clarity.

Thank you for your suggestion. We have adjusted the line styles and line widths in Figure 11 and modified the line widths in Figure 10 to improve clarity.

Page 25, Line 518: "However, when measuring their refractive indices in the laboratory, it is …". Refractive indices of aerosol particles are not usually directly measured in the laboratory, but are derived from absorption or extinction measurements assuming an optical model.

Apologies for the oversight. We have updated the word 'measuring' to 'determining'.

Page 25, Line 530: "Under and ideal scenario, where no instrumental defects needed to be corrected, the look-up tables for the homogeneous super-spheroid models were able to fit the measurements at any size.". There are a few issues here: First, the size and scattering truncation corrections are not "defects". A nephelometer measuring scattering in an angular range from 7 to 170 degrees is not a defect. Extrapolation (or correction) is the procedure used to obtain the total integrated scattering. Second, I would not refer to an "ideal scenario" as a case where beta_scat(0 to180) is approximated by beta_scat(7 to 170). Was a solution found using the super-spheroid model fitted with an approximation beta_scat(7 to 170)? What does that say about the consistency of the model?

Thank you for your feedback. We have revised the term 'defects' to 'unavoidable technical limitations' to accurately reflect the nature of the challenges faced in the measurements. Regarding the clarification about case E1, we apologize for the confusion. In case E1, the target scattering coefficients for the inversion models in retrieval are indeed the same as those

of the baseline case (beta_scat(0 to 180)). The case E2 refers to the numerical simulations with scattering truncation correction. The conversion from beta_scat(7 to 170) to beta_scat(0 to 180) leads to slight inconsistency between the homogeneous super-spheroid model and the baseline case in E2.

Page 26, Line 552: "The retrieved refractive indices were found to be size-dependent. As the size increased, the imaginary part decreased.". Where is this shown in the authors analysis? How do you explain this result if you have assumed the same composition in each simulation? Was that observed for both the homogeneous and inhomogeneous models?

The discrepancies in the size distribution mainly contribute to this observation, as illustrated in Figures 7 and 9 in the manuscript. Notably, this phenomenon is specific to homogeneous models, which utilize a single refractive index for the entire model, whereas inhomogeneous models incorporate different refractive indices for different components. The refractive indices assumed for the inhomogeneous models are considered to be independent of size (see Sect. 2.2). A comprehensive explanation is provided in Sect. 3.2.

"However, it should be noted that the refractive indices obtained at small sizes may not be applicable to large sizes as they depend on the sizes in the retrieval process when assuming a homogeneous model (Figure 9). The size-dependent refractive indices are more evident in E3/E4 than in E1/E2 due to significant discrepancies in size distribution between the baseline case and the inversion models. Specifically, the imaginary parts decrease as the size increases, as shown in Figure 7 and Figure 9. The variation of imaginary parts with particle size is less evident when the inversion models share an identical size distribution with the baseline case. However, the variation of imaginary parts is more pronounced at 355 nm in E1 compared with other wavelengths, primarily due to the particle's relatively larger size parameter at 355 nm, which leads to variations in optical properties with the size parameter."

Page 26, Line 556: Please define "true values". Are the authors referring to the complex refractive indices obtained with the inhomogeneous super-sphere model? Why are they not shown?

The term "true values" refers to the optical properties of the baseline case, which specifically pertains to the inhomogeneous super-spheroid model. Only the optical properties of the baseline case (extinction coefficients, SSA, asymmetry factor, phase matrix) can be compared. It is important to note that the baseline case does not utilize a single refractive index for the entire model, which is a characteristic exclusive to homogeneous models.

---

## Author Comment (AC2)

In this response letter, comments from the reviewer are highlighted in black font, our responses are in blue font, and modifications made to the manuscript are indicated in red font.

**Reviewer #2**

General comments.

This manuscript describes a method to apply the super-spheroid model to simulated aerosol particles and investigate their non-sphericity. To this aim, the authors draw on their previous work and assess the uncertainties introduced by model-dependent data interpretation. In this case, they could explain their contribution to aerosol science more clearly. The authors use the super-spheroid model to mimic experimental data, then proceed to interpret such data with a look-up table (LUT) generated with the same model while comparing the results with those obtained with a LUT based on the spherical model. Although mentioned in the introduction, other numerical models are not considered for data analysis. I am not sure how generalizable the results of this work actually are. The main weaknesses I see in the work are that it uses the same model as an input and an interpretation framework and that it does not directly confront the spheroidal model. Even though the spherical model is the benchmark for interpreting experimental data, going one step further would make this simulation work more complete. Overall, I am afraid that several (key) parts of the manuscript are not clear enough and would suggest revising its written form. I would also suggest, to improve readability, revising the use of the past tense (it is sometimes hard to tell what precedes this work and what is part of it) and the typesetting of equations and variables. I address the manuscript's sections with some comments followed by a selection of notes on specific lines.

Thank you for your comprehensive feedback. We appreciate your insights and suggestions for improving the clarity and completeness of the manuscript. We have made improvements based on your feedback:

1.  The spheroid-based LUT:

We selected the inhomogeneous super-spheroid model as the baseline case to represent realistic dust aerosols in laboratory experiments. The shape of this model exhibits comparable non-sphericity with actual dust aerosols, as depicted in Figure 12 of the revised manuscript. We incorporated the homogeneous super-spheroid LUT because it shares the same shape as the baseline case. This inclusion helps in understanding the uncertainties when inversion models align with the target (dust aerosols). Furthermore, we introduced other numerical models-based LUTs, such as the spheroid-based LUT, in Section 3.4 to facilitate deeper investigation and comprehensive discussion.

2.  Tense and Typesetting Revisions:

We revised the use of past tense and adopted present tense to describe the numerical simulations and results in our work. Additionally, we adjusted the typesetting of variables for improved readability. For example, 'RI1' has been changed to 'RI(super-spheroid)' and 'RI2' to 'RI(sphere)'.

3.  Abstract and Summary Enhancement:

We have revised the abstract and summary to make them more effective and concise, ensuring they accurately reflect the key findings and contributions of our study.

Abstract.

I believe it is longer than it needs to be while not serving its purpose quite effectively. After the first few introductory lines, it reads more like an excerpt of a discussion section than a concise synopsis of the authors' work. A key point is that the manuscript is about numerical rather than experimental results, which is not clear from either the abstract or the title.

Thank you for your feedback. We appreciate your insight into the length and effectiveness of the manuscript. We have endeavored to ensure that the abstract provides a concise and clear synopsis of our numerical research findings.

The abstract has been revised as follows:

"Mineral dust particles are nonspherical and inhomogeneous; however, they are often simplified as homogeneous spherical particles for retrieving the refractive indices from laboratory measurements of scattering and absorption coefficients. The retrieved refractive indices are then employed for computing the optical properties of spherical or nonspherical dust model particles with downstream applications. This study theoretically investigates uncertainties involved in the aforementioned rationale based on numerical simulations and focuses on a wavelength range of 355 to 1064 nm. Initially, the optical properties of nonspherical and inhomogeneous dust aerosols are computed as baseline cases. Subsequently, the scattering and absorption coefficients of homogeneous spheres and super-spheroids are computed at various refractive indices and compared with those of inhomogeneous dust aerosols to determine the dust refractive index. To mimic the real laboratory measurement, the size distribution of the baseline case is assumed to be unknown and determined through a process akin to using optical particle counters for sizing. The resulting size distribution differs from the original one of the baseline cases. The impact of discrepancies in size distributions on retrieving the dust refractive index is also investigated. We found that these discrepancies affect scattering and absorption coefficients, presenting challenges in accurately determining the refractive index, particularly for the real parts. Additionally, the retrieved refractive indices are noted to vary with particle size primarily due to differences in size distribution, with imaginary parts decreasing as the particle size increases. A comparison between sphere and super-spheroid models shows that the former tends to underestimate the imaginary parts, leading to an overestimation of single scattering albedo. This study underscores the importance of employing consistent nonspherical models for both refractive index retrieval and subsequent optical simulation in downstream applications. Nevertheless, the impact of refractive index uncertainties on asymmetry factor and phase matrix is found to be minimal, with particle shape playing a more significant role than differences in the imaginary parts of the dust refractive index."

lines 15) If the authors want to elaborate on the 'realistic' and 'ideal' scenarios in the abstract, a brief explanation of these terms should be given.

Thank you for your suggestion. We have decided to omit the references to the 'realistic' and 'ideal' scenarios in the abstract. The corresponding statements have been revised accordingly.

line 17) discrepancies among which distributions?

The discrepancies in size distribution between the inversion models and the baseline case. We have revised the statements.

"…The resulting size distribution differs from the original one of the baseline cases. The impact of discrepancies in size distributions on retrieving the dust refractive index is also investigated."

line 23) why should they?

It means that the uncertainties in the imaginary part and single scattering albedo in the numerical simulations is not higher than 0.002 (0.0007) and 0.03 (0.01), respectively, under conditions of high (low) absorption.

We have refrained from using these expressions in the revised abstract to avoid potential misinterpretation of the results. Thank you for bringing this to our attention.

line 25) I am not sure it fits in the abstract, but the authors should discuss somewhere what they consider a large variation in the asymmetry factor.

The significant variation in the asymmetry factor can be attributed to the variations in the real parts of the refractive indices. However, it is important to note that the disparity in the asymmetry factor between the sphere model and the super-spheroid model is primarily caused by differences in shape. We have removed the sentence about the variation from the abstract.

In line 539 of the revised manuscript:

"Significant variations are observed in the asymmetry factor at wavelengths of 865 nm and 1064 nm for M-E1 and M-E3 (Figure 10). These variations are attributed to the variations in the real parts of the refractive indices. To simplify the discussion, the refractive indices retrieved from the homogeneous super-spheroid models are referred to as RI(super-spheroid), while those from the homogeneous sphere models are referred to as RI(sphere). Below 865 nm, the real parts of RI(sphere) are set to the default value of 1.52 for M-E1, and the same is done for RI(super-spheroid) for M-E3 as the target values deviate significantly from the values in the look-up table. However, at 865 nm and 1064 nm, the target values fall within the look-up table, and the extinction coefficients are well matched in M-E1 and M-E3. Despite this, the retrieved real parts deviate significantly from the value of 1.52."

line 26) I am afraid this sentence may be misleading as it can be understood only after having read the whole article.

Thank you for your feedback. We have made revisions to the sentence accordingly.

Introduction.

Overall, the contextualization of the models mentioned here could be improved. There are too few references about the state of the art the authors are motivated to expand upon.

Thank you for your valuable suggestions. We acknowledge the importance of providing sufficient references to the state of the art to enhance the comprehensiveness of the contextualization. We have revised this section based on the comments below.

When reading the central body of this section, many references point to the limitations of the spheroid model applied to experimental data: one could expect this to be the main motivation of this work. Yet the matter is not elaborated further. The authors should state more clearly

what they mean by 'revisiting the laboratory measurements' using a numerical method. Since they are producing simulated data, they can control every parameter involved in the study, therefore, they can check the assumptions they make as to why some results are inconsistent.

Thank you for your insightful feedback on this section. It is indeed crucial for us to clarify the main motivation behind our work. In revisiting the laboratory measurements using a numerical method, we aim to explore the implications of inversion models possessing the same shape as the target particles and conduct a thorough theoretical examination of uncertainties arising from principles in laboratory measurements of mineral dust refractive indices at short wavelengths. Furthermore, we aim to investigate the consistency in optical properties between realistic dust aerosols and homogeneous models. We have articulated these points more clearly in the revised version of the manuscript.

line 36) please shortly define the 'super-spheroid model' and the rationale behind its introduction. The super-spheroid model requires more parameters than the spherical model, therefore, it can better fit experimental data by design. Its advantages should be better argued, given the aim of this work.

Thank you for your suggestion. We have modified the expression:

"Among these models, the spheroid model is commonly used in remote sensing applications. To improve upon the spheroid model, the super-spheroid model, which extends the dimensions of both the sphere and spheroid models, has been proposed to provide a more comprehensive framework for describing the shape of dust particles. Initial studies have shown that homogeneous super-spheroid dust models align well with laboratory measurements… "

line 45) I assume the authors mean the volume-equivalent diameter, but many definitions of this parameter are possible.

Yes, the geometric diameter here means the volume-equivalent diameter. We have added a parenthesis to make this clear.

[revised manuscript text omitted]

Experimental design.

The fact that this work does not involve actual measurements but consists of numerical calculations is a defining feature. This paper should, in my opinion, state its scope more clearly. I am fully persuaded of the importance of simulative research, which is why I would advise against using the name "dust sample" or "measurement" for what is a numerical model. While I share the authors' point of view, I'm afraid that obtaining experimental data with the instruments they mention involves more than just truncating the integration interval or tuning some parameters to address a non-ideal scenario. I would go into greater detail about the

assumptions the authors make to generate the population of particles they then proceed to study.

Thank you for your insightful feedback. We have provided further clarification regarding the scope and nature of the numerical simulations in this study. We refrained from using terms like "dust sample" or "measurement" to accurately reflect the nature of our simulations. Additionally, we have included a statement at the beginning of this section to clearly emphasize that this study consists solely of numerical calculations and does not involve laboratory measurements. Moreover, we have revised the introduction to the procedures in the numerical simulations to provide detailed explanations of the assumptions underlying our numerical models and how we generate the particle population for study.

The authors might consider moving lines 156–175 at the beginning of the 'Experimental Design' section.

Thank you for your suggestion. This section provides a detailed description of the inhomogeneous super-spheroid models, including the composition and the refractive indices for each component. We believe it is appropriate to include these details in this section (2.2 Model and computational method) as they contribute to a comprehensive understanding of the models.

line 99) please explain what 'inhomogeneous' stands for.

We have added a short description for the inhomogeneous model.

"The inhomogeneous super-spheroid dust models, internally mixed with several minerals (see Sect. 2.2), are considered as the baseline case, mimicking the dust samples used in the laboratory experiments."

line 118, 230–235) I believe that the process of changing or correcting the size distribution requires some more explanation.

We have implemented modifications to enhance the clarity of the size distribution conversions.

"The size conversions between the sphere models and the super-spheroid models can be expressed as shown below:

$$\frac{dN}{dD_{geo}} = \frac{dN}{dD_{geo,sphere}} \cdot \frac{dD_{geo,sphere}}{dD_{geo}} = \frac{dN}{dD_{geo,sphere}} \cdot \left( \frac{1}{C_f} - \frac{D_{geo} \cdot \frac{dC_f}{dD_{geo}}}{C_f^{\,2}} \right), \qquad (3)$$

in which $D_{geo} = D_{geo,sphere} \cdot C_f$ , representing the geometrical diameter for the inhomogeneous or homogeneous super-spheroid models. Then, the size distributions for the super-spheroid models ($dN/dD_{geo}$) (in Error! Reference source not found. b) are derived from the size distributions for the sphere models ($dN/dD_{geo,sphere}$) using Eq. (3). For a specific size parameter, the $D_{geo}$ for super-spheroids are smaller than those for spheres.

Different from E3/E4 scenarios, the size distribution for all models (sphere, homogeneous and inhomogeneous super-spheroids) in E1 and E2 are assumed to the same. The size distribution in E1 and E2 are…"

line 124) here the authors refer to unavoidable technical limitations that are easily taken care of rather than defects. On the other hand, modeling possible stochastic or systematic experimental errors would be surely interesting.

Thank you for the clarification. We have revised the term 'defects' to 'unavoidable technical limitations'. In this study, we assumed a relative uncertainty of 8% in the scattering coefficients and 30% in the absorption coefficients. While modeling possible stochastic or systematic experimental errors is intriguing, we did not consider stochastic experimental errors as our main focus is to assess the assumption of spherical models for dust aerosols.

figure 1) I am not sure how fair it is to compare results from the spherical-based LUT with those from the super-spheroid LUT since the input dust itself is generated with the super-spheroidal model. Alternatively, an ellipsoid-based LUT would be interesting.

Thank you for your suggestion. We have utilized the super-spheroid LUT because the super-spheroid model shares the identical shape with the baseline case, aiding in understanding the uncertainties if the inversion models align with the target (dust aerosols). Additionally, we have included a spheroid-based LUT in Sect. 3.4 to analyze the results as the inversion model varies.

line 137) why were these specific aspect ratios (geometries) chosen?

We observed that equiprobable aspect ratios ranging from 0.5 to 2 are appropriate for fitting the measurements of dust aerosols using the super-spheroid models. However, conducting light scattering calculations for the inhomogeneous super-spheroid model is exceedingly time-consuming and demands substantial computational resources. Consequently, due to these resource constraints, we restricted our selection to three aspect ratios for the super-spheroid models. Our investigation revealed that models employing these chosen aspect ratios yield optical properties comparable to those derived from models utilizing equiprobable aspect ratios (see Figure R*1*). To elucidate our rationale, we have provided additional explanatory remarks regarding the selection of these aspect ratios in the manuscript.

"Additionally, three aspect ratios (0.5, 1.0, 2.0) are considered. Although we have observed that equiprobable aspect ratios ranging from 0.5 to 2 are appropriate for fitting the measurements of dust aerosols using the super-spheroid models (Lin et al., 2018), we restrict our selection to three aspect ratios (0.5, 1.0, 2.0) for the super-spheroid models due to computational resource constraints. Nonetheless, models employing these chosen aspect ratios yield optical properties comparable to those derived from models utilizing equiprobable aspect ratios."

[Figure]

Figure R1. The phase matrices (with integration on the size distribution) of the homogeneous super-spheroid models with various aspect ratios.

line 159) please provide some references supporting this statement and some further comments on how the refractive index distributions were set.

We have added the references and reworded this statement.

"We choose the sample from Algeria for our study because it has a medium iron content, which closely approximates the mean values of the global average (Di Biagio et al., 2019; Go et al., 2022)."

For the inhomogeneous models, the refractive indices for different minerals were allocated to distinct parts of the models (Wang et al., 2023). We believe the refractive index distributions you mentioned pertain to the refractive index range utilized for retrieval. We have included comments elaborating on why the refractive index range was set to 1.40-1.70 for the real part and 0.0001 to 0.015 for the imaginary part, as suggested.

"In the optical database for homogeneous sphere and super-spheroid models, the real part of the refractive index ranges from 1.40 to 1.70 at intervals of 0.01, while the imaginary part varies from 0.0001 to 0.015 at steps of 0.0001. These values are determined based on literature values of refractive indices (Di Biagio et al., 2019 and references therein)."

line 197) the geometric diameter should be defined earlier in the text.

Thank you for your suggestions. We have provided an explanation within parentheses for the geometric diameter at its initial usage.

"…optical particle counters could underestimate the geometric diameter (volume-equivalent diameter) of dust aerosols…"

line 204) the retrieval of the size distribution is a crucial step of data inversion and should be considered as a factor influencing final results. Given the numerical nature of the work, this could be investigated.

We agree with that. The conversion from the optical diameter obtained by OPC to the geometric diameter is crucial and necessary. The statements about the experiments in the manuscript may be unclear and misleading. In this study, the experiment E1 represents the situation that the geometric size of the inhomogeneous model can be accurately obtained. Then the homogeneous super-spheroid model and sphere model use the same size distribution as the inhomogeneous model. While the experiment E3 represents the situation that the size of inhomogeneous model can not be obtained directly, but is measured by the OPC. The optical diameters obtained by OPC are further converted into the geometric diameters of the homogeneous super-spheroid models and the sphere models. Therefore, the differences in the results in E1 and E3 help us to investigate the impact of the retrieval of the size distribution on the retrieved refractive indices.

For clarity, we also have added a short description of the size correction in the section 2.1 Overall procedure.

"In the numerical simulations, we have incorporated two correction processes based on the actual laboratory experiments. The first correction is the size correction, which is employed to determine the geometric size of the particles from imaginary OPC measurements…The second correction is the scattering truncation correction, which is associated with the unavoidable technical limitations in measurements of scattering coefficients."

line 216) please describe more in depth the observed biases.

We have added more discussion about the bias in retrieving the size distribution and the paragraph has been rewritten.

"The $C_f$ values of the inhomogeneous models are about 8% lower than those of the homogeneous models. This suggests that the bias in retrieving size distribution for inhomogeneous and irregular particles is not only caused by the difference in model shapes, but also by the imperfect representation of inhomogeneity. According to Eq. (2), any disparity in the microphysical properties (e.g., shape, absorptivity) will result in a difference in $I_{sca}$, ultimately leading to a bias in $C_f$ values. However, the variation in model shapes leads to a dominant bias when using sphere models for retrieval. The use of incorrect refractive indices for the conversion of size can introduce biases in the converted particle size when comparing homogeneous and inhomogeneous models. Prior values of refractive indices are crucial for accurate size conversion. Sensitivity studies have shown that the conversions are much less sensitive to n than to k. The change in absorptivity results in significant variations in the scattering cross section, as dust aerosols exhibit moderate absorption in the visible band. The first guess values of $k$ is essential. We consider the literature values in this study (Di Biagio et al., 2019). However, the literature value is not the sole option, the first guess values of refractive indices can be optimized through an iterative retrieval. The retrieval can start with a rough guess value and then obtain a more precise value. The retrieved values can be used as new prior values, and the retrieval can be repeated until optimal values are obtained. To maintain consistency with the laboratory studies (Di Biagio et al., 2017b, 2019; Ryder et al., 2013), we do not consider this iterative retrieval to obtain the size distribution. The conversion factors for homogeneous models and inhomogeneous models are typically similar if they have comparable absorptivity. For instance, the inhomogeneous super-spheroid models showed similar trends to the low-absorbing homogeneous super-spheroid models in which $k = 0.001$ (**Error! Reference**

**source not found.**a, b).”

Results and discussion.

I would be more cautious about the generality of the conclusions that may be drawn from this work, having its design in mind. It is certainly valuable to provide some links to experimental work by adapting the premises and parameters of the simulations, yet I wouldn't say the E4 case covers them completely. The authors state that defining the size of irregular particles necessarily leads to discrepancies. I see it is a critical step but it lacks the insights it deserves. With all the input parameters being known (unlike field measurements), the analysis could be more aware and detailed than it is at present. This comes to mind when they attribute the observed discrepancies to differences in the size distribution, for instance.

Thank you for your valuable feedback. We acknowledge that the E4 case may not fully represent laboratory experiments. Our numerical simulations offer insights into how uncertainties arise and their potential extent, serving as a reference for experimental studies. We have revised Section 3.4 and conducted additional investigations to provide more detailed interpretations of the observed discrepancies, particularly regarding differences in size distribution. For detailed information, please refer to the revised manuscript.

line 316) the authors point out the limited range of the LUT more than once. Is this a critical limitation? If so, would it be feasible to extend it, even if it were to include unlikely values of the refractive index or size?

We highlighted that the scattering and absorption coefficients of the baseline case (target values) exceeded the range of the look-up tables for the inversion models to underscore the significantly large bias. We chose not to extend the LUT, as doing so would have minimal impact in reducing this substantial bias, and the computational cost for the light scattering calculation of nonspherical particles is high. Figure R2 depicts the original LUT and the extended one created by the sphere models with real parts ranging from 1.20 to 1.70. Despite the extension, the target values remained uncovered by the LUT. The bias is primarily attributed to the discrepancy in size distributions.

[Figure]

Figure R2. The original LUT and extended LUT (the real part) created by the sphere models. In the original LUT, the real part of the refractive index ranges from 1.40 to 1.70 at intervals of 0.01, while the imaginary part varies from 0.0001 to 0.015 at steps of 0.0001. In contrast, the expanded LUT features the real part of the refractive index ranging

from 1.20 to 1.70 at intervals of 0.01, while the imaginary part remains unchanged.

line 318) why are the uncertainties this large?

The uncertainty of the scattering coefficients is set at 8% in this study, as indicated in Sect. 3.1 (Lines 290-291 in the original manuscript). Nevertheless, as particle size increases, the scattering coefficients exhibit reduced sensitivity to variations in the real parts of refractive indices. Consequently, when compared with the range of scattering coefficients in the look-up tables, the uncertainty (8%) appears disproportionately large.

We have added an explanation of why the overall dimensions of the look-up tables decrease with increasing size at the beginning of Sect. 3.1.

"**Error! Reference source not found.** illustrates the scattering and absorption coefficients of the baseline case (target values), as well as the look-up tables for the super-spheroid models and sphere models in the simulation scenarios of E1 and E2. It is noteworthy that the overall dimensions of the look-up tables diminish with increasing size. As the particle size increases, the scattering coefficients become less sensitive to changes in the real parts of the refractive indices. This phenomenon can be explained by the optical theorem, which states that the extinction cross sections(including scattering and absorption cross sections) are approximately twice the geometric projected area as the size increases, regardless of the refractive indices (Liou, 2002). However, the absorption coefficients are significantly influenced by the imaginary parts, which in turn affect the scattering coefficients."

lines 322 and ff) it might be redundant to mention a model that gives even larger discrepancies, I would rather try to quantify the extent to which the definition of size could affect the results.

Thank you for your suggestion. In the manuscript, we used the geometric diameter (volume-equivalent diameter) to describe the size of sphere models and super-spheroid models. The optical properties between these models were generally similar at small sizes, resulting in close retrieved refractive indices. We also mentioned the effective radius (diameter) as a size descriptor for non-spherical particles. The selection of a size descriptor was crucial for size correction and optical property calculation. According to Saito and Yang (2022), the effective radius is the most suitable size descriptor for non-spherical particles. Therefore, we verified the results using the effective radius. However, the effective radius calculated for the super-spheroid model was significantly smaller than the volume-equivalent radius. The optical properties of sphere models calculated using the effective radius were smaller than those of super-spheroids. This suggests that the geometric diameter is a better size descriptor for retrieving refractive indices using sphere models.

Figure R3 illustrates that look-up tables for sphere models do not cover the 'truth value' at small sizes (S) when the effective radius is used. For size L (Figure R4), the differences between the sphere models and the actual values are even greater. It is challenging to accurately quantify this effect because the discrepancies vary with the size of the models and wavelengths. Therefore, we have presented a range of 50-70% for this. The original expression (lines 322 and onwards) has been corrected as follows:

"However, we find that the discrepancies are even larger when the effective radius is used. The effective radius is smaller than the geometric radius at the same size parameter for the super-spheroid model. As a result, the simulated scattering coefficients of the sphere models using the effective radius are smaller than those using the geometric radius. The retrieval fails even

at small size (S), and the difference of scattering coefficients between those calculated by sphere models and the baseline case can be 50-70% depending on the size and wavelength. Therefore, it is believed that the geometric diameter is better suited for retrieval in this study."

[Figure]

Figure R3. The look-up tables for refractive indices produced by the sphere model and super-spheroid model for size S by using the effective radius as the size descriptor. The markers and symbols are similar to the Figure 5 in the manuscript.

[Figure]

Figure R4. Similar to Figure R3, but for size L.

lines 336–350) given the lack of literature on the refractive index of goethite and its prominent role in determining the refractive index of the ensemble of particles, why did the authors include it in the simulations? How do the results change if they don't?

We include the refractive index of goethite into the simulations for two primary reasons:

1. Since our study relies on numerical simulations without direct comparison to experimental data, the accuracy of the refractive indices of minerals has minimal impact on our analysis. The refractive index of goethite is used as reference values in our simulations.

2. Furthermore, we seek to assess whether the refractive indices of magnetite can serve as a substitute for goethite at wavelengths below 460 nm and above 700 nm. Previous studies lacked refractive index data for goethite within these wavelength ranges, prompting us to explore alternative solutions. Our findings indicate a potential overestimation of the imaginary parts of goethite beyond 700 nm when assuming the imaginary parts of magnetite.

Excluding goethite from the simulations would result in a comparable absorptivity of the baseline case at 863 nm and 1064 nm compared to that at 633 nm. Moreover, the retrieved imaginary parts of the refractive indices would decrease from 355 nm to 633 nm and remain relatively constant until 1064 nm.

line 368 and ff.) the spherical model and the super-spheroid model give very close results, particularly considering the error bars. Do the authors have any hypotheses or explanations as

to this matter?

We have provided additional explanations for this phenomenon.

"Therefore, retrieving the imaginary parts solely from the absorption coefficients exhibits reduced sensitivity to the model shape when identical size distributions are utilized. This is because the extinction coefficients are primarily influenced by particle sizes, and the calculated absorption coefficients are similar for models with the same size distributions. As a result, the target absorption coefficient is mapped to similar imaginary parts in the look-up tables for inversion models with the same size distribution."

line 384-389) the argument being made here could be clearer, I believe it's important to go into greater detail about such differences. Since these are simulations, these parameters are under control and are available for analysis.

Thank you for your suggestion. We have elaborated on this aspect in the revised manuscript.

"Similar to Figure 5, the target values and the look-up tables in E3 and E4 are illustrated in Figure 8. Note that a significant discrepancy emerges between the baseline case and the homogeneous super-spheroid models as the size increases, which is inconsistent with the findings in Figure 5. Furthermore, the discrepancy for the sphere models is even larger. This discrepancy can be attributed to the differences in size distributions. These differences are not influenced by the size descriptor of the non-spherical particle but are directly caused by the discrepancies in the optical properties between the baseline case and the models when using the OPC to measure the size of individual particles.

As described in Eq. (2), the OPC measures the scattering intensity of individual particles with a metric that is influenced by both particle size and optical properties. When using the OPC for particle sizing, differences in optical properties, influenced by shape and inhomogeneity, between the baseline case and the inversion models result in biases in particle size estimation across different models. Despite the homogeneous super-spheroid models having identical shapes to the baseline case, differences associated with the inhomogeneity introduce size biases between them. Additionally, for sphere models, deviations in shape from the baseline case, combined with inhomogeneity, further contribute to significant discrepancies in particle size estimation. Therefore, accurately retrieving $n$ is challenging because the scattering coefficients are highly sensitive to the size distribution."

Besides, we have provided a more detailed discussion in Sect. 3.4. Please refer to the revised manuscript for details.

line 390) I think this is a central point that should be investigated further, especially because of their great effect on optical properties.

Thank you for your suggestion. We have conducted additional investigation on this matter in Section 3.4. Please refer to the revised manuscript for details.

lines 410 and ff) this appears to be a significant limitation of the method: what are these very strict conditions required to accurately retrieve n?

The strict conditions imply the absence of discrepancies in the size distribution and morphology

between the baseline case and the inversion models, as demonstrated in numerical simulation E1. However, it is essential to note that successful retrieval does not guarantee that the inversion model shares identical optical properties with the baseline case (refer to Sect. 3.3).

line 455) this is one of the main weaknesses of the spherical approximation. How does the phase function of super-spheroids compare to laboratory measurements? Some information about how they were calculated would be helpful (e.g. possible rotational averages).

Thank you for confirming. The revised manuscript now includes the information about considering random orientations on Line 173. "We calculate the single particle optical properties using the IITM method for inhomogeneous super-spheroid models, considering random orientations…"

The phase functions of super-spheroids are in good agreement with laboratory measurements, as reported in a previous study (Lin et al., 2018).

line 459) although I share the authors' concern, it should be noted that considerable progress has been made since the literature cited here was first published.

Thank you for your suggestion. We have revised this sentence and included additional references.

"The discrepancy in the asymmetry factor may introduce a significant bias in climate modelling. Despite notable advancements, many climate models still rely on sphere models to simulate dust aerosols (Balkanski et al., 2007; Danabasoglu et al., 2020; Hess et al., 1998; Hurrell et al., 2013; Liu et al., 2016; Mishchenko et al., 1995)."

line 461) 'simplify'?

Corrected.

line 462) 'are'?

Corrected.

line 485) please expand the caption of Figure 11 with the information included in the main text.

Thanks; we have made the necessary correction.

line 508) I would recommend that the reasons for these difficulties be probed deeper and discussed in this section more thoroughly.

Thank you for your suggestion. We have expanded on this point in the revised manuscript, providing more detailed discussion in this section. Please refer to the revised manuscript for details.

Summary.

I appreciate the purpose of this section, also in light of the length of the manuscript. I wonder if it would not be more effective to move its content partly into the introduction and partly into the discussion.

Thank you for your suggestion. We have revised this section to focus primarily on the main findings, removing some introductory details about how the numerical simulations were conducted. Please refer to the revised manuscript for details.

References.

I find the bibliography skewed toward less recent results in some sections of the text. Some updates would be beneficial, specifically because of the rapid scientific advances in the field of simulations. It would also be worth double-checking how relevant some of the self-citations are to the discussion and the points being made by the authors.

Thank you for your feedback. We have revised the bibliography according to your suggestions and added more recent references to reflect the latest developments in the field of simulations. Additionally, we have reviewed the self-citations to ensure their relevance to the discussion and the points being made in the manuscript.

**References**

[revised manuscript text omitted]

---

## Author Response (AR2)

In this response letter, comments from the reviewer are highlighted in black font, our responses are in blue font, and modifications made to the manuscript are indicated in red font.

We have uploaded data to https://zenodo.org/records/11093920 and added a few sentences to the data availability section.

"The data used in the numerical simulations are available at https://zenodo.org/records/11093920. The additional data from this study are available on request."

**Report #1**

**Submitted on 16 Apr 2024**

**Anonymous referee #2**

I think that the authors have addressed many critical points in their manuscript and that it is now much clearer. I list some minor comments below while recommending a thorough proofreading of the manuscript.

Thank you very much for your positive feedback and suggestions. We have made revisions accordingly and conducted a thorough proofreading. We would like to express our sincere appreciation and extend special thanks to the two anonymous reviewers for their valuable contributions to improving this manuscript. We have added a few sentences to the acknowledgment section of our manuscript.

"…We would like to thank the two anonymous reviewers for their invaluable help in improving the manuscript."

Figure 1 (line 157): in the caption, the authors might consider adding a sentence like 'see section ... for details' if they feel that an explanation of the flowchart here would not be concise enough.

Thank you for your suggestion. We have added a sentence in the caption.

"Target scattering coefficient denotes the scattering coefficient of the baseline case. See section 2.3 for more details."

Line 221 and throughout the manuscript: the subscripts that are labels and not indices (such as 'sca', 'min', or 'geo'), the function sin, and the differential symbol should be typeset upright.

Thank you for your suggestion. We have made adjustments to those subsicripts and functions.

Line 235: Note that?

Corrected.

Line 235: there is a missing space before µm.
Corrected.

Figure 7 (line 435): I would suggest including a marker for the Bouguer–Lambert lines like in the other models.
Thank you for your suggestion regarding Figure 7. In this figure, the Bouguer–Lambert results serve only as a reference for comparison, and we want to highlight the contrast between the sphere and super-spheroid models. After some careful considerations, we decided not to use markers for the Bouguer–Lambert lines.

Line 355 and throughout the manuscript: please make sure that the imaginary part of the refractive index and other symbols referring to physical observables are consistently typeset in italics.
According to your suggestions, we have gone through and revised the manuscript, ensuring that the imaginary part of the refractive index and other symbols indicating physical observables are consistently typeset in italics.

Figure 6 (line 420): to improve readability, I would consider including a marker where data points are, in addition to the lines in the plots.
Thank you for your suggestion. We have now added markers to the data points along with the plot lines, as you suggested.

Line 441: please include a comment about the nan values in the table caption and in the related section.
Thank you for your suggestion. We have added a comment about the nan values in the caption and related section.
Line 414 "In cases where target values exceed the range of the look-up table, "nan" values are provided in Table 3 (the same in Table 4)."
Line 441 (the table caption): "The nan values indicate that target values fall outside the look-up table."

**Report #2**

**Submitted on 21 Apr 2024**

**Anonymous referee #1**

1) The article has been significantly improved since the first version.

We greatly appreciate your feedback and suggestions. We would like to express our sincere appreciation and extend special thanks to the two anonymous reviewers for their valuable contributions to improving this manuscript. We have added a few sentences to the acknowledgment section of our manuscript.

"…We would like to thank the two anonymous reviewers for their invaluable help in improving the manuscript."

2) The language and readability of the abstract could be improved to more clearly explain the objectives, methodology, and conclusions of the article.

Thank you for your suggestion. We have revised the abstract a little bit for better readability.

3) The authors called the last section "Summary". Should it be changed to "Conclusions"?

Thank you for your suggestion. We have changed the title to "Conclusions".